# Beyond VLM-Based Rewards: Diffusion-Native Latent Reward Modeling

Gongye Liu [1]  Bo Yang [1]  Yida Zhi [1]  Zhizhou Zhong [1]  Lei Ke [2]  Didan Deng [3]  Han Gao [3]  Yongxiang Huang [3]
Kaihao Zhang [4]  Hongbo Fu [1]  Wenhan Luo [✉ 1]

## Abstract

Preference optimization for diffusion and flow-matching models relies on reward functions that are both discriminatively robust and computationally efficient. Vision-Language Models (VLMs) have emerged as the primary reward provider, leveraging their rich multimodal priors to guide alignment. However, their computation and memory cost can be substantial, and optimizing a latent diffusion generator through a pixel-space reward introduces a domain mismatch that complicates alignment. In this paper, we propose **DiNa-LRM** , a **di**ffusion-**na**tive **l**atent **r**eward **m**odel that formulates preference learning directly on noisy diffusion states. Our method introduces a noise-calibrated Thurstone likelihood with diffusion-noise-dependent uncertainty. DiNa-LRM leverages a pretrained latent diffusion backbone with a timestep-conditioned reward head, and supports inference-time noise ensembling, providing a diffusion-native mechanism for test-time scaling and robust rewarding. Across image alignment benchmarks, DiNa-LRM substantially outperforms existing diffusion-based reward baselines and achieves performance competitive with state-of-the-art VLMs at a fraction of the computational cost. In preference optimization, we demonstrate that DiNa-LRM improves preference optimization dynamics, enabling faster and more resource-efficient model alignment. Our source code is available at https://github.com/HKUST-C4G/diffusion-rm.

## 1. Introduction

Diffusion (Ho et al., 2020) and flow-matching models (Liu et al., 2023) have become the dominant paradigm for high-quality visual generation, achieving remarkable progress in image (Esser et al., 2024; Wu et al., 2025a) and video synthesis (Brooks et al., 2024; Wan et al., 2025). As these models scale, aligning their outputs with human preferences has emerged as a critical challenge. A range of preference alignment algorithms, including Reward Feedback Learning (Xu et al., 2023), Direct Preference Optimization (Wallace et al., 2024; Yang et al., 2024; Liu et al., 2025b), and RL-based approaches such as GRPO (Liu et al., 2025a; Xue et al., 2025b), have been proposed and shown strong empirical results on large-scale foundation models (Wu et al., 2025a; Seedream et al., 2025). Across these methods, the reward model is a key bottleneck because it provides the supervision signal that steers optimization.

Most existing reward models for visual generation are built on large multimodal models, and recent work has increasingly adopted vision-language models (VLMs) as reward backbones (Liu et al., 2025b; Ma et al., 2025; Wang et al., 2025c). Compared to earlier CLIP-based rewards (Xu et al., 2023; Wu et al., 2023a; Kirstain et al., 2023), VLM-based rewards substantially improve accuracy and robustness. This trend is further amplified by backbone scaling, as larger VLMs often provide more robust discriminative ability after large-scale pretraining (Wu et al., 2025b). Meanwhile, using VLM rewards in alignment can be costly, especially when reward evaluation is queried repeatedly during optimization. Moreover, VLM rewards typically operate in pixel space, whereas latent diffusion generators are trained and optimized in latent space (Rombach et al., 2022), which introduces a latent-to-pixel mismatch that complicates alignment and increases system overhead, especially for reward-gradient methods (Xu et al., 2023; Prabhudesai et al., 2023).

These limitations motivate revisiting reward modeling from a diffusion-native perspective. Diffusion models form another family of large-scale pretrained backbones (Esser et al., 2024; Wan et al., 2025), and prior work shows that their generative pretraining yields rich representations that transfer to discriminative objectives, ranging from classification (Li et al., 2023; Xiang et al., 2023) to adversarial discrimination (Sauer et al., 2024). This raises a natural question: *can diffusion backbones be turned into general-purpose reward models that remain competitive in preference discrimination while being more optimization-friendly for alignment?*

[1]The Hong Kong University of Science and Technology [2]Tsinghua University [3]Huawei Hong Kong AI Framework & Data Technologies Lab [4]The Australian National University. Correspondence to: Wenhan Luo <whluo@ust.hk>.

*Proceedings of the 43rd International Conference on Machine Learning*, Seoul, South Korea. PMLR 306, 2026. Copyright 2026 by the author(s).

Recent attempts (Zhang et al., 2025b; Mi et al., 2025) have begun to explore diffusion models as noise-aware rewards under specific alignment paradigms, often focusing on preference optimization over noisy intermediate states. While promising, this line of work is often tied to particular training algorithms and does not directly study diffusion backbones as general-purpose reward models under the same usage scenario as VLM rewards. In contrast, we focus on the reward model itself and study diffusion backbones as reusable reward models under the same usage scenario as VLM rewards, namely, scoring clean samples via latent-space evaluation. Our goal is to unlock the intrinsic discriminative capability of large-scale diffusion pretraining through diffusion-native preference modeling.

To this end, we propose **DiNa-LRM** , a diffusion-native latent reward model. DiNa-LRM formulates preference learning directly on noisy diffusion states by extending the Thurstone model with a noise-calibrated comparison uncertainty that scales with the diffusion noise level. Built on a pretrained latent diffusion backbone, DiNa-LRM predicts timestep-aware rewards in the VAE latent space with a scoring head. At inference time, DiNa-LRM supports noise ensembling that aggregates evidence from multiple timesteps within the reward head, providing a diffusion-native test-time scaling knob for more robust scoring. Experiments show that DiNa-LRM substantially improves over diffusion-based reward baselines and narrows the gap to strong VLM-based rewards on benchmark evaluations, while reducing memory overhead and improving optimization dynamics in preference optimization under the same setup.

Our contributions are summarized as follows:

- **Diffusion-native preference formulation.** We extend the Thurstone preference model from clean samples to noisy diffusion states by introducing a noise-calibrated comparison uncertainty that scales with the noise level.

- **Inference-time scaling via noise ensembling.** We build a timestep-aware latent reward model on top of a pretrained latent diffusion backbone, and propose inference-time noise ensembling that aggregates multi-timestep features.

- **Empirical evaluation of alignment behavior.** We demonstrate that diffusion-native rewards substantially improve over diffusion-based baselines and narrow the gap to strong VLM rewards on benchmarks.

## 2. Related Works

**CLIP-based Reward Models** Early reward modeling (Wu et al., 2023b; Xu et al., 2023; Kirstain et al., 2023; Wu et al., 2023a; Zhang et al., 2024; Liang et al., 2024; Ye et al., 2026b) for text-to-image generation often repurposes CLIP-style vision-language pretraining as a proxy

for human preference. Representative methods such as PickScore (Kirstain et al., 2023), HPS-v2 (Wu et al., 2023a), and ImageReward (Xu et al., 2023) fine-tune CLIP (Radford et al., 2021) or BLIP (Li et al., 2022) on human preference datasets to predict a single scalar score. Subsequent works have extended this paradigm to multi-dimensional assessments (Zhang et al., 2024) and fine-grained feedback signals (Liang et al., 2024). While computationally efficient, these models typically couple a frozen or fine-tuned encoder with a lightweight MLP head. Consequently, their performance is inherently bounded by the representational capacity of the pretrained CLIP models.

**VLM-based Reward Models** With rapid advances in large vision-language models (Hurst et al., 2024; Wang et al., 2024; Bai et al., 2025), recent reward modeling has increasingly shifted toward stronger VLM backbones that support richer semantic understanding, improved robustness, and better generalization. A common practice is to replace the VLM's language head with a regression (He et al., 2024; Liu et al., 2025b; Ma et al., 2025) or logit-based head (Wu et al., 2025b; Gong et al., 2025; Zhang et al., 2025a), and optimize the model for Bradley–Terry (Bradley & Terry, 1952) or MSE objectives. Emerging generative reward models further leverage chain-of-thought (Wang et al., 2026; Wu et al., 2025c), thinking-with-image (Wang et al., 2025b), and VLM-as-a-verifier (Zhang et al., 2025c) strategies to produce structured reasoning before scoring. Despite their effectiveness, VLM-based reward models typically operate in pixel space and incur high inference cost. Crucially, their reliance on discrete text generation often renders gradient propagation impractical, limiting their application in on-policy, reward-gradient-based alignment.

**Diffusion Models for Discriminative Task** Beyond generation, recent studies have demonstrated that diffusion backbones learn transferable representations suitable for discriminative objectives such as classification (Li et al., 2023; Xiang et al., 2023). Diffusion models have also been explored as discriminators (Sauer et al., 2024; Yin et al., 2024; Lu et al., 2025a) in adversarial training settings, leveraging their ability to process noisy inputs and their compatibility with latent-space pipelines. These findings suggest that diffusion backbones are natural candidates for reward modeling. Concurrent efforts (Zhang et al., 2025b; Mi et al., 2025) have investigated diffusion-based reward modeling, primarily focusing on step-level rewards over noisy intermediate states to facilitate trajectory-level optimization. In contrast, our work targets diffusion-based reward modeling in general-purpose preference alignment, where reward evaluation on clean images or latents. And we further investigate diffusion-native designs that improve reward quality and optimization efficiency under this setting.

## 3. Preliminaries

**Diffusion Models**   In the discrete-time setting, diffusion models (Ho et al., 2020; Song et al., 2021) define a forward noising process that gradually perturbs a data sample $\boldsymbol{x}_0 \sim p_{\text{data}}$ into a noisy state $\boldsymbol{x}_t$ over $T$ steps:

$$\boldsymbol{x}_t = \alpha(t)\boldsymbol{x}_0 + \sigma(t)\boldsymbol{x}_T, \qquad \boldsymbol{x}_T \sim \mathcal{N}(0, \boldsymbol{I}), \quad (1)$$

where $\{\alpha(t), \sigma(t)\}$ is a predefined noise schedule (typically satisfying $\alpha^2(t) + \sigma^2(t) = 1$). The reverse denoising process is parameterized by a neural backbone, such as U-Net (Rombach et al., 2022) or DiT (Peebles & Xie, 2023), and is commonly trained via $\epsilon$-prediction that conditioned on timestep $t$ and an optional condition $\boldsymbol{c}$:

$$\mathcal{L}_{\text{DM}}(\theta) = \mathbb{E}_{\boldsymbol{x}_0, \boldsymbol{x}_T, t, \boldsymbol{c}} \left[ \|\boldsymbol{x}_T - \epsilon_\theta(\boldsymbol{x}_t, t, \boldsymbol{c})\|_2^2 \right]. \quad (2)$$

**Flow Matching Models**   Similarly, flow-matching models (Lipman et al., 2023; Liu et al., 2023) generate samples by solving a continuous-time transport ODE, which can be described using the same linear forward form as Equation 1, with $\alpha(t) = 1 - t$ and $\sigma(t) = t$ defined in the rectified flow framework. The model learns a time-dependent velocity field $v_\theta(\boldsymbol{x}_t, t, \boldsymbol{c})$ via $v$-prediction regression,

$$\mathcal{L}_{\text{FM}}(\theta) = \mathbb{E}_{\boldsymbol{x}_0, \boldsymbol{x}_1, t, \boldsymbol{c}} \left[ \|v_\theta(\boldsymbol{x}_t, t, \boldsymbol{c}) - (\boldsymbol{x}_1 - \boldsymbol{x}_0)\|_2^2 \right]. \quad (3)$$

**Latent Space Modeling**   Diffusion and flow-matching share a similar formulation (Lipman et al., 2024), so the proposed methods could apply to both. Throughout this paper, **we adopt flow-matching notation**: $\boldsymbol{x}_0$ denotes clean data and $\boldsymbol{x}_1$ denotes Gaussian noise. Both models typically operate in the frozen VAE latent $\mathcal{Z}$ (Rombach et al., 2022), with optimization restricted to the denoising backbone.

## 4. Method

In this section, we propose **DiNa-LRM**, a **Di**ffusion-**Na**tive **L**atent **R**eward **M**odel. We first introduce a diffusion-native preference formulation by extending the Thurstone model from clean samples to noisy diffusion states, where comparison uncertainty is explicitly calibrated by the noise level. We also present the fidelity-loss objective and timestep sampling strategies for optimization. We then describe the latent reward architecture built on a pretrained diffusion backbone with a query-based scoring head. Finally, we present an inference-time noise ensembling algorithm that aggregates multi-timestep features for test-time scaling. An overview of the full framework is provided in Figure 1.

### 4.1. Diffusion-native Preference Formulation

Given a text prompt $\boldsymbol{c}$ and a preference label indicating which sample is better within a pair $(\boldsymbol{x}_0^+, \boldsymbol{x}_0^-)$, our goal is to learn a scalar scoring function $r_\theta(\boldsymbol{x}_0, \boldsymbol{c}) \in \mathbb{R}$ that assigns higher scores to preferred samples.

**Thurstone Model on Clean Samples**   Human preference annotations are inherently noisy, since subjective judgments can be ambiguous and labeling inconsistent. Following the Thurstone model's formulation (Thurstone, 2017), we treat human preferences as noisy observations derived from an underlying reward function $r_\theta(\boldsymbol{x}_0, \boldsymbol{c})$. Specifically, we model the perceived quality of a sample as a stochastic reward $u(\boldsymbol{x}_0, \boldsymbol{c})$, defined as the additive composition of a deterministic score and a Gaussian noise term:

$$u(\boldsymbol{x}_0, \boldsymbol{c}) = r_\theta(\boldsymbol{x}_0, \boldsymbol{c}) + \eta, \qquad \eta \sim \mathcal{N}(0, \sigma_u^2), \quad (4)$$

where the noise term $\eta \sim \mathcal{N}(0, \sigma_u^2)$ captures the intrinsic ambiguity in human judgment. For a labeled pair $(\boldsymbol{x}_0^+, \boldsymbol{x}_0^-)$, the preference probability is formulated by:

$$\mathbb{P}(\boldsymbol{x}_0^+ \succ \boldsymbol{x}_0^- \mid \boldsymbol{c}) = \Phi\left( \frac{r_\theta(\boldsymbol{x}_0^+, \boldsymbol{c}) - r_\theta(\boldsymbol{x}_0^-, \boldsymbol{c})}{\sqrt{\sigma_u^2 + \sigma_u^2}} \right), \quad (5)$$

where $\Phi(\cdot)$ denotes the CDF of the standard normal distribution. This framework allows us to learn the reward parameters $r_\theta$ by maximizing the likelihood of the observed preference data.

**Noise-calibrated Thurstone**   Equation 5 provides a straightforward way to learn $r_\theta(\boldsymbol{x}_0, \boldsymbol{c})$ from clean preference samples. However, diffusion and flow-matching backbones are pretrained to process *noisy* states $\boldsymbol{x}_t$ rather than clean samples $\boldsymbol{x}_0$. To bridge this distributional gap, we propose **noise-calibrated Thurstone**, which extends preference learning to the noisy samples $(\boldsymbol{x}_t^+, \boldsymbol{x}_t^-)$.

Given a timestep $t$, noisy states are formed by the standard forward noising process (Equation 1),

$$\boldsymbol{x}_t = \alpha(t)\boldsymbol{x}_0 + \sigma(t)\boldsymbol{\epsilon}, \qquad \boldsymbol{\epsilon} \sim \mathcal{N}(0, \boldsymbol{I}), \quad (6)$$

where $\sigma(t)$ controls the noise magnitude. Intuitively, a larger $t$ removes more semantic information from the underlying sample, making preference judgments on $\boldsymbol{x}_t$ less certain. We formalize this effect by modulating the comparison uncertainty $\sigma_u(t)$ as a function of the diffusion noise level $\sigma^2(t)$ in Equation 6:

$$\sigma_u^2(t) = k\,\sigma^2(t) + \sigma_u^2, \quad (7)$$

where $\sigma_u$ represents the baseline ambiguity on clean samples in Equation 4, and $k$ scales the growth of uncertainty. In our paper, we empirically set $k = 2$ and $\sigma_u = 0.1$.

Consequently, the preference likelihood on noisy states is:

$$\mathbb{P}(\boldsymbol{x}_t^+ \succ \boldsymbol{x}_t^- \mid t, \boldsymbol{c}) = \Phi\left( \frac{r_\theta(\boldsymbol{x}_t^+, t, \boldsymbol{c}) - r_\theta(\boldsymbol{x}_t^-, t, \boldsymbol{c})}{\sqrt{\sigma_u^2(t) + \sigma_u^2(t)}} \right), \quad (8)$$

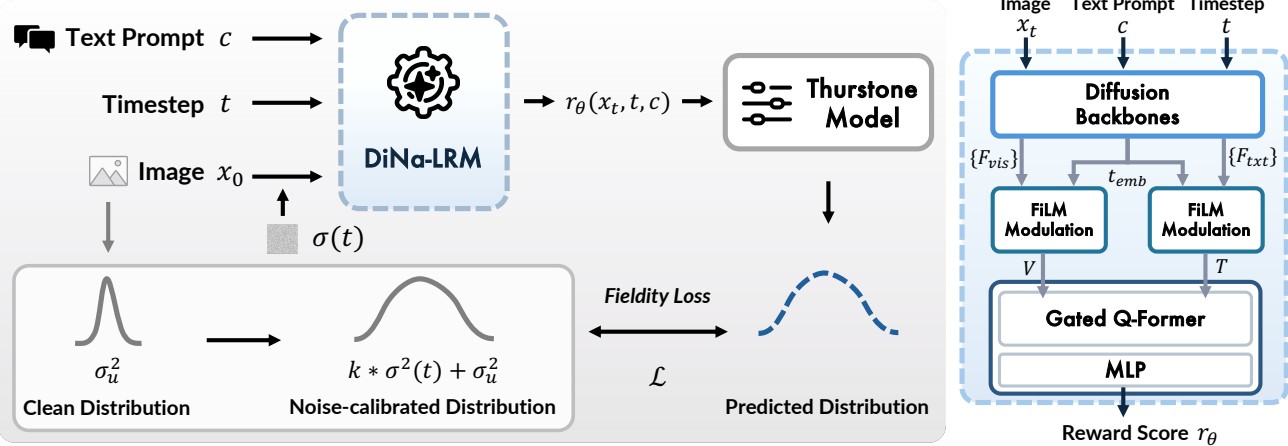

*Figure 1.* **Overview of DiNa-LRM . Left: Diffusion-native Preference Learning.** During training, clean preference pairs $(x_0^+, x_0^-, c)$ are perturbed to noisy states $(x_t^+, x_t^-)$ and evaluated by a time-conditioned reward model $r_\theta$. We employ a noise-calibrated Thurstone likelihood where comparison variance scales with the diffusion noise level $\sigma_t$, and optimize via a fidelity loss $\mathcal{L}$. **Right: Latent Reward Architecture.** Multi-layer visual and text features extracted from a latent diffusion backbone are FiLM-modulated by timestep embeddings $t_{emb}$. These features are aggregated through a gated Q-Former and an MLP to produce a scalar reward score.

where $r_\theta(\boldsymbol{x}_t, t, \boldsymbol{c})$ denotes a time-aware reward model that takes the noisy input together with its noise level. This formulation treats preference learning as a family of objectives indexed by $t$, where the supervision signal remains constant while the confidence is regularized by the noise level.

This noise-calibrated approach offers several advantages for diffusion-based reward modeling. First, **Distributional Alignment**. By defining the reward objective over $\boldsymbol{x}_t$, we ensure the model's input distribution remains consistent with the noisy manifold encountered during diffusion pre-training. Second, **Inference Versatility**. The time-dependent formulation $r_\theta(\boldsymbol{x}_t, t, \boldsymbol{c})$ enables flexible reward estimation during deployment. To approximate the preference for a clean sample, one can either evaluate the model at a fixed low-noise index or aggregate predictions across the noise schedule via ensembling. This allows for sophisticated inference-time scaling strategies that leverage the global trajectory rather than a single point estimate. Third, **Uncertainty-Aware Regularization**. The noise-calibrated variance $\sigma_u^2(t)$ explicitly accounts for the diminishing semantic signal-to-noise ratio as $t$ increases. By inducing a conservative preference likelihood in high-noise regimes, this mechanism prevents uninformative, high-variance gradients from destabilizing training, effectively acting as a noise-aware formulation for preference learning.

**Training** Following prior work (Chen et al., 2025), we adopt fidelity loss (Tsai et al., 2007) for optimization. Let $y \in \{0, 1\}$ denote the preference label for a clean pair $(\boldsymbol{x}_0^+, \boldsymbol{x}_0^-)$ (with $y = 1$ indicating $\boldsymbol{x}_0^+ \succ \boldsymbol{x}_0^-$), we minimize:

$$\mathcal{L}_{\text{fid}}(\theta) = \mathbb{E}_{(\boldsymbol{x}_0^+, \boldsymbol{x}_0^-, \boldsymbol{c}, y), t \sim q(t)} \left[ 1 - \sqrt{y\,\hat{p}_\theta + (1 - y)\left(1 - \hat{p}_\theta\right)} \right]. \tag{9}$$

The timestep distribution $q(t)$ determines the noise level used for preference learning. We consider three strategies: **(i) Fixed**, using a single timestep $t = t^\star$, when $t^\star = 0$, the preference formulation reduces to Equation 5. **(ii) Uniform**, sampling $t$ uniformly from $\mathcal{U}(0, 1)$; and **(iii) Logit-Normal** (Esser et al., 2024), which samples timesteps by drawing a Gaussian $\mathcal{N}(\mu, \sigma^2)$ in logit space and mapping it to $(0, 1)$ via a sigmoid. In practice, we find that fixed sampling is sensitive to $t^\star$ and less robust, while uniform and logit-normal yield comparable results. We adopt *uniform* by default for simplicity, and provide a more detailed comparison in Figure 5.3.

### 4.2. Latent Reward Architecture

Our reward model takes a noisy latent $\boldsymbol{x}_t$ together with prompt $\boldsymbol{c}$ and timestep $t$, and predicts a scalar reward $r_\theta(\boldsymbol{x}_t, t, \boldsymbol{c})$. It is instantiated on top of a pretrained latent diffusion backbone (SD3.5-Medium (Esser et al., 2024) in most experiments) and a query-based reward head. **All computations are performed in VAE latent space, with the VAE kept frozen.**

**Multi-layer diffusion features.** Given $(\boldsymbol{x}_t, t, \boldsymbol{c})$, the diffusion backbone produces hidden-state sequences at multiple layers for both visual and textual streams (if available). In practice, extracting features from *a small subset* of layers $\mathcal{S}$ is sufficient, and we do not require all layers of the backbone. We formulate it as:

$$\left\{ \mathbf{F}_{\text{vis},t}^{(i)}, \mathbf{F}_{\text{txt},t}^{(i)} \right\}_{i \in \mathcal{S}} = \text{Backbone}(\boldsymbol{x}_t, t, \boldsymbol{c}), \tag{10}$$

where $\mathbf{F}_{\text{vis},t}^{(i)} \in \mathbb{R}^{N_v \times C}$ and $\mathbf{F}_{\text{txt},t}^{(i)} \in \mathbb{R}^{N_t \times C}$ denote the visual and text token features extracted at layer $i$, respectively.

**Timestep-conditioned adaptation.** To make the reward head explicitly aware of the noise level, we apply FiLM-style modulation (Perez et al., 2018) to each selected layer feature using the timestep embedding $t_{\text{emb}}$. Each adapted feature is projected to a lower-dimensional subspace, concatenated across layers, and linearly fused into unified token sequences: $\mathbf{V_t} \in \mathbb{R}^{N_v \times d}$ for visual tokens and $\mathbf{T_t} \in \mathbb{R}^{N_t \times d}$ for text tokens.

**Q-Former Scoring** We aggregate $\mathbf{V_t}$ and $\mathbf{T_t}$ using a query transformer with $N_q$ learnable query tokens. The queries attend to visual and text tokens via value-gated (Qiu et al., 2025) cross-attention. We then refine the queries with a second visual-only cross-attention block, followed by a lightweight FFN. Finally, we map the pooled query representation to a scalar reward:

$$r_\theta(\boldsymbol{x}_t, t, \boldsymbol{c}) = \text{MLP}\Big(\text{Pool}(\tilde{\mathbf{Q}})\Big) \in \mathbb{R}, \qquad (11)$$

where $\tilde{\mathbf{Q}}$ denotes the final query states and $\text{Pool}(\cdot)$ is mean pooling over queries. The query-based head also naturally supports multi-noise ensembling: features extracted at multiple timesteps can be adapted and concatenated as a longer context token sequence, and the same Q-Former head can be applied once to produce an aggregated score.

### 4.3. Inference-Time Scaling via Noise Ensembling

Our reward model is defined on noisy inputs and predicts $r_\theta(\boldsymbol{x}_t, t, \boldsymbol{c})$. At inference time, however, the goal is typically to assign a reliable score to a clean sample $(\boldsymbol{x}_0, \boldsymbol{c})$. A simple choice is to evaluate the reward at a fixed low-noise level:

$$\hat{r}(\boldsymbol{x}_0, \boldsymbol{c}) = r_\theta(\boldsymbol{x}_{t^\star}, t^\star, \boldsymbol{c}). \qquad (12)$$

Beyond a single noise level, diffusion models provide multiple noise-conditioned "views" of the same sample and thus admit a natural test-time scaling knob. To reduce sensitivity to a single evaluation noise level and improve the performance, we evaluate the sample at multiple timesteps $\{t_k\}_{k=1}^K$ and aggregate them **within the reward head**. Specifically, we extract backbone features at each $(\boldsymbol{x}_{t_k}, t_k, \boldsymbol{c})$, apply the timestep-conditioned modulation, and concatenate the resulting context tokens across timesteps:

$$\mathbf{V}_{\text{ensemble}} = \text{Concat}\big(\mathbf{V}_{t_1}, \dots, \mathbf{V}_{t_K}\big) \in \mathbb{R}^{(K \times N_v) \times C},$$
$$\mathbf{T}_{\text{ensemble}} = \text{Concat}\big(\mathbf{T}_{t_1}, \dots, \mathbf{T}_{t_K}\big) \in \mathbb{R}^{(K \times N_v) \times C}. \qquad (13)$$

After that the same query-based scoring head is applied once to produce an aggregated score $\hat{r}(\boldsymbol{x}_0, \boldsymbol{c})$.

This token-level ensembling serves as a diffusion-native form of inference-time scaling. We observe that the reward model's performance varies across datasets and evaluation timesteps, indicating that different timesteps may emphasize different aspects of the underlying representation. By combining features from different timesteps, the model can leverage complementary evidence and produce a more stable score, at the cost of additional inference compute.

## 5. Experiments

### 5.1. Experimental Setup

**Training Details** We adopt SD3.5-Medium (Esser et al., 2024) as the diffusion backbone. Our proposed DiNa-LRM is trained on a valid subset of HPDv3 (Ma et al., 2025), which contains approximately 0.8M pairwise comparisons. Training is conducted for one epoch on 8 GPUs with 80G VRAMs. We adopt AdamW with weight decay $0.01$, a constant learning rate of $5 \times 10^{-5}$, and a total batch size of $64$. We maintain an exponential moving average (EMA) of model weights with decay $0.995$ throughout training, and use the EMA weights for evaluation.

Following standard diffusion-training practice (Podell et al., 2024), we adopt bucketed training for variable-resolution images. Each sample is resized to its nearest resolution bucket, and over 80% of images preserve their original resolution and aspect ratio after bucketing. For timestep sampling, we use the *Uniform* strategy, drawing $t \sim \mathcal{U}(0, 1)$. For each preference pair, the chosen and rejected samples are perturbed with the same noise to form $(\boldsymbol{x}_t^+, \boldsymbol{x}_t^-)$. More implementation details are provided in Appendix E.

**Evaluation Protocol** We compare against representative reward models from three categories: *(i) CLIP-base Reward Models:* ImageReward (Xu et al., 2023), PickScore (Kirstain et al., 2023), HPSv2 (Wu et al., 2023a), and MPS (Zhang et al., 2024); *(ii) VLM-based Reward Models:* UnifiedReward (Wang et al., 2025c), UnifiedReward-CoT (Wang et al., 2026), and HPSv3 (Ma et al., 2025); *(iii) Diffusion-based Reward Models:* LRM-SD1.5 (Zhang et al., 2025b) and LRM-SDXL.

Following prior works (Ma et al., 2025), we evaluate on a diverse suite of test sets, including ImageReward (Xu et al., 2023), HPDv2 (Wu et al., 2023a), HPDv3 (Ma et al., 2025), and GenAI-Bench (Jiang et al., 2024), to assess generalization across data quality and annotation sources. For all datasets, we report *pairwise preference accuracy*, i.e., the fraction of comparisons where the reward model assigns a higher score to the preferred sample. Unless otherwise specified, DiNa-LRM is evaluated using a single noise level with $t = 0.4$. For test-time ensembling, we compute rewards at three noise levels $t \in \{0.2, 0.5, 0.7\}$ and aggregate them as described in subsection 4.3. The ensemble set is chosen to span low-, mid-, and high-noise regimes, balancing robustness and computational overhead.

*Table 1.* **Pairwise Preference Accuracy across different benchmarks.** We compare CLIP-/VLM-/DIFFUSION-based reward models. Within each group, the best result is shown in **bold**. DiNa-LRM * indicates that we adopt token-level multi-noise inference scaling.

| Model | Backbone | Pairwise Preference Accuracy (%) | | | | |
|---|---|---|---|---|---|---|
| | | ImageReward | HPDv2 | HPDv3 | GenAI Bench | Avg |
| *CLIP-base RM* | | | | | | |
| ImageReward (Xu et al., 2023) | CLIP | 65.15 | 73.95 | 58.74 | 63.41 | 65.31 |
| PickScore (Kirstain et al., 2023) | CLIP | 62.73 | 79.44 | **65.67** | **69.98** | 69.45 |
| HPSv2 (Wu et al., 2023a) | CLIP | 65.62 | 82.58 | 64.69 | 67.62 | 70.13 |
| MPS (Zhang et al., 2024) | CLIP | **66.37** | **83.27** | 64.33 | 68.08 | **70.51** |
| *VLM-base RM* | | | | | | |
| UnifiedReward (Wang et al., 2025c) | LLaVA-OV-7B | 63.82 | 83.10 | 71.96 | **72.38** | 72.81 |
| UnifiedReward-Think (Wang et al., 2026) | LLaVA-OV-7B | 58.54 | 82.70 | 66.07 | 70.91 | 69.56 |
| HPSv3 (Ma et al., 2025) | Qwen2VL-7B | **67.03** | **85.36** | **76.03** | 70.95 | **74.84** |
| *Diffusion-based RM* | | | | | | |
| LRM-SD1.5 (Zhang et al., 2025b) | SD-1.5 | 59.17 | 72.39 | **54.05** | 60.86 | 61.62 |
| LRM-SDXL (Zhang et al., 2025b) | SDXL | 60.35 | 71.19 | 53.80 | 61.58 | 61.73 |
| **DiNa-LRM (Ours)** | SD3.5-M-2B | 60.34 | 82.13 | **75.04** | 68.43 | 71.49 |
| **DiNa-LRM * (Ours)** | SD3.5-M-2B | **61.75** | **84.31** | 74.86 | **68.98** | **72.48** |

## 5.2. Reward Modeling

The preference prediction accuracy on various test sets is presented in Table 1. Overall, DiNa-LRM substantially improves over prior diffusion-based reward models, while remaining slightly behind the strongest VLM-based judge. This suggests that diffusion features from a modern foundation backbone can serve as a considerably stronger basis for preference prediction. Across datasets, HPSv3 attains the highest average accuracy, while DiNa-LRM demonstrates that diffusion-native rewards in latent space can achieve competitive accuracy with clear optimization advantages. Importantly, latent-space reward evaluation avoids pixel-space overhead and latent-to-pixel mismatch, making it more efficient during alignment.

Additionally, test-time multi-noise ensembling provides a consistent but modest improvement. Aggregating rewards across multiple noise levels improves the average accuracy from 71.49% to 72.48%. This supports our motivation that different noise levels expose complementary evidence in diffusion representations, and ensembling can reduce sensitivity to the evaluation timestep.

We further provided cross-backbone evaluations and comparisons in Appendix C.1 and Appendix C.3. Additional results on SDXL, FLUX.1-Dev, and Z-Image-Turbo show that the proposed formulation is not specific to SD3.5-M.

## 5.3. Ablation Studies

**Inference-time Noise Level.** We analyze sensitivity to the inference-time noise level $t$. Figure 2 reports the average accuracy as a function of $t$ for models trained with different timestep schedules. Preference accuracy is maximized at intermediate noise levels, with the best performance consistently attained around $t \in [0.3, 0.7]$. In contrast, evaluating at *near-clean states* ($t = 0$) or *highly noisy states* ($t = 0.8$) leads to degraded accuracy. A plausible explanation is that intermediate-noise representations better balance semantic fidelity and discriminative signal. We therefore use $t = 0.4$ as the default single-noise inference setting.

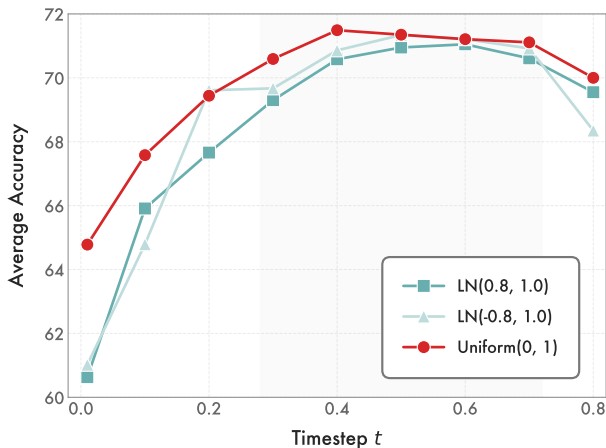

*Figure 2.* **Effect of Inference-time Noise Level. Uniform** sampling performs consistently well across a wide range of $t$, with accuracy peaking at mid-range timesteps.

*Table 2.* **Ablations on timestep schedule and uncertainty modeling.** We vary the training timestep schedule (CONST/UNIFORM/LOGITNORMAL) and the Thurstone variance modeling (FIXED vs. NOISE-CALIBRATED). **Ensemble** indicates multi-timestep ensembling is enabled. **Bold**: Best Performance. Underline: Second Best.

| Timestep Schedule | Variance | Ensemble | Pairwise Preference Accuracy (%) | | | | |
|---|---|---|---|---|---|---|---|
| | | | ImageReward | HPDv2 | HPDv3 | GenAI Bench | Avg |
| *Constant timestep (single-noise training)* | | | | | | | |
| $t = 0$ | Fixed | | 58.61 | 59.20 | 74.37 | 67.55 | 64.93 |
| $t = 0.2$ | Fixed | | 59.59 | 62.92 | 74.67 | 67.97 | 66.29 |
| $t = 0.7$ | Fixed | | 59.94 | 73.75 | 73.28 | 68.02 | 68.75 |
| *Uniform schedule ($t \sim \mathcal{U}(0,1)$)* | | | | | | | |
| Uniform | Fixed | | 60.88 | 78.72 | 75.11 | 68.01 | 70.68 |
| Uniform | Fixed | ✓ | 60.77 | 78.16 | 74.68 | 68.01 | 70.41 |
| Uniform (**Ours**) | Noise-Calibrated | | 60.34 | 82.13 | 75.04 | 68.43 | 71.49 |
| Uniform (**Ours**) | Noise-Calibrated | ✓ | **61.75** | **84.31** | 74.86 | **68.98** | **72.48** |
| *LogitNormal schedule* | | | | | | | |
| $LN(0.8, 1.0)$ | Noise-Calibrated | | 60.36 | 78.73 | 74.89 | 68.35 | 70.58 |
| $LN(0.8, 1.0)$ | Noise-Calibrated | ✓ | 60.88 | 80.31 | 74.68 | 68.86 | 71.18 |
| $LN(-0.8, 1.0)$ | Noise-Calibrated | | 60.42 | 79.08 | **75.40** | 68.52 | 70.86 |
| $LN(-0.8, 1.0)$ | Noise-Calibrated | ✓ | 61.44 | 80.76 | 75.07 | 68.43 | 71.43 |

**Timestep Schedules** We first study how the training-time timestep schedule affects diffusion-native reward learning. We compare fixed-noise training at a single timestep (Const $t \in \{0, 0.2, 0.7\}$) against distributional schedules that expose the model to a range of noise conditions. As summarized in Table 2, fixed-noise training can achieve reasonable performance on the in-domain set like HPDv3, but generalizes poorly to other test sets, leading to substantially lower average accuracy. In contrast, distributional schedules markedly improve out-of-domain generalization, indicating that *covering multiple noise regimes during training is important for robust preference learning*. Among distributional schedules, Uniform sampling provides a strong and stable default. We further consider LogitNormal schedules (following the SD3.5 training recipe (Esser et al., 2024)) to explicitly bias training toward higher-noise (LogitNormal(0.8, 1.0)) or lower-noise (LogitNormal(−0.8, 1.0)) regions. Results show that Logit-Normal is competitive, yet Uniform yields higher average accuracy overall in our setting. Overall, the performance differences between distributional schedules are moderate, with Uniform providing the best results in our experiments.

**Noise-Calibrated Variance** We evaluate whether explicitly tying the comparison variance to the diffusion noise level in the Thurstone model improves preference learning. We compare a constant-variance Thurstone variant (**Fixed**, $\sigma_u = 0.5$) against the proposed **Noise-Calibrated Variance**. As shown in Table 2, noise calibration yields clear gains in overall performance, with the most pronounced improvements on out-of-domain benchmarks and under multi-noise ensembling. In particular, Noise-Calibrated substantially boosts HPDv2 accuracy (e.g., $78.72 \rightarrow 82.13$ for single inference and $78.16 \rightarrow 84.31$ with ensembling), and improves the averaged accuracy more significantly when combined with ensembling ($70.41 \rightarrow 72.48$). We hypothesize that noise-dependent uncertainty modeling provides a better-calibrated training signal across noise regimes, encouraging the model to *learn more diverse and complementary features at different noise levels*. This diversity is especially beneficial for robustness across different datasets and for inference-time scaling via multi-noise ensembling.

**Backbone Adaptation.** We compare LoRA fine-tuning on the backbone against training only the reward head with the backbone frozen. As reported in Appendix 3, LoRA adaptation provides consistent gains, while the frozen-backbone variant remains competitive. This suggests that pretrained diffusion representations already provide useful signals for preference learning, and lightweight backbone adaptation can further refine these features toward the preference objective. Additional architectural ablations are provided in Appendix C.2. We compare the gated Q-Former reward head with simpler aggregation strategies and further vary the number of extracted diffusion layers. These results show that the diffusion-native formulation remains effective under simpler heads, while query-based selective fusion and richer multi-layer features provide stronger average performance.

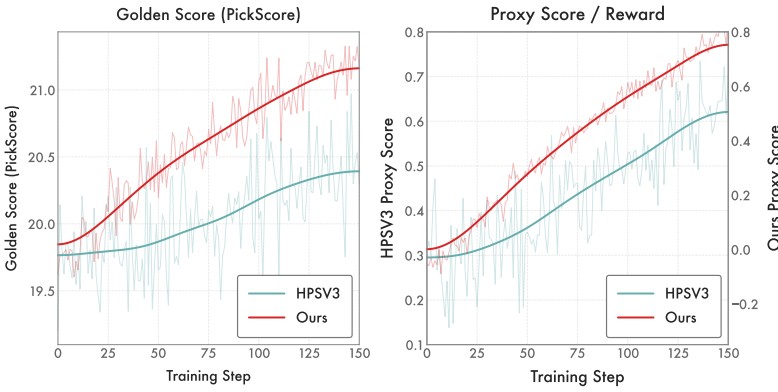

Figure 3. **Training Curves (ReFL on SD3.5-M)**. We optimize with either HPSv3 or DiNa-LRM (Ours) as the **proxy** reward. We report the optimized proxy score *(right)* and an external held-out **golden metric** *(PickScore; left)*. DiNa-LRM improves the proxy score faster while the golden metric increases in tandem.

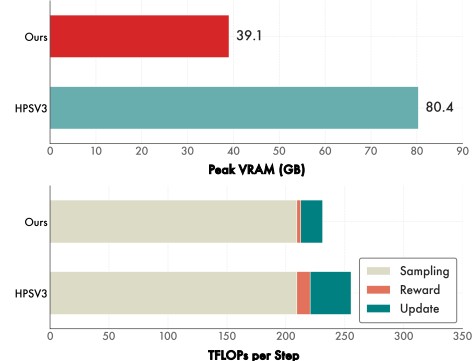

Figure 4. **Efficiency Analysis**. **Peak VRAM** *(top)* and **per-step TFLOPS** *(bottom)* for a single ReFL optimization step is reported.

Table 3. **Effect of backbone adaptation.**

| Training Strategy | HPDv3 | GenAI-Bench | Avg (4 sets) |
|---|---|---|---|
| Freeze backbone | 73.52 | 67.09 | 70.27 |
| LoRA fine-tuning | 75.04 | 68.43 | 71.49 |

**Multi-Noise Ensembling.** Beyond a single noise level, diffusion models provide multiple noise-conditioned views of the same sample, and the reward performance can benefit from aggregating features across timesteps. As shown in Table 1 and Table 2, when combined with our noise-calibrated variance, token-level multi-noise ensembling generally improves the overall performance, with particularly notable gains on out-of-domain benchmarks, suggesting that different timesteps capture complementary discriminative features that enhance robustness across different datasets.

### 5.4. Preference Alignment

To assess the practical utility of DiNa-LRM , we evaluate its performance in the context of post-training alignment. We adopt ReFL (Xu et al., 2023) as a representative reward-gradient algorithm, where the generative model is optimized by backpropagating gradients through the reward signal. We compare our diffusion-native reward against HPSv3, a state-of-the-art VLM-based reward model.

**Experimental Setup** We optimize SD3.5-M on the Pick-a-Pic (Kirstain et al., 2023) dataset. To ensure a controlled comparison, we maintain identical ReFL implementations and optimization hyperparameters across all experiments, varying only the source of the rewards. We run 150 optimization steps with a batch size of 256. For DiNa-LRM , a clean latent sample $(x_0, c)$ is scored via a fixed low-noise evaluation $r(x_0, c) \triangleq r_\theta(x_{0.4}, 0.4, c)$. In contrast, HPSv3 is computed on the decoded image in pixel space.

**Optimization Dynamics** Inspired by Wang et al. (2025a), we designate the optimization objective as a *proxy score* and track a held-out *golden score* (PickScore) that is withheld from the optimization. This allows us to track alignment progress while detecting potential reward hacking. As shown in Figure 3, optimizing with our diffusion-native reward yields accelerated improvement in the proxy score compared to the HPSv3 baseline, and the golden score increases in tandem with the proxy reward, showing no obvious evidence of reward hacking at the early iterations. Conversely, HPSv3 demonstrates slower convergence in both proxy and golden scores under the same training steps.

**Efficiency Analysis** We quantify the efficiency gains of our diffusion-native approach by profiling single-step ReFL updates at $1024 \times 1024$ resolution with a batch size of 1, reporting peak VRAM and FLOPs. As summarized in Figure 4, our latent reward is substantially cheaper to optimize. It reduces peak memory by $51.4\%$, reward-calculation FLOPs by $71.1\%$, and optimization-phase FLOPs by $46.4\%$ relative to HPSv3. These gains stem from computing rewards natively in the VAE latent space, which avoids expensive decoding and reduces overhead during alignment.

These findings demonstrate that diffusion-native rewards provide an effective and optimization-friendly reward signal for post-training alignment, improving training dynamics while substantially reducing the resource cost. We further validate this conclusion in broader alignment settings in the supplementary materials: Appendix C.1 reports a more detailed comparison against LRM-SDXL under ReFL settings, Appendix C.3 studies transfer from an SD3.5-M reward model to SD3.5-L alignment within a shared latent space, and Appendix C.4 extends the evaluation to Flow-GRPO-Fast online RL.

# 6. Conclusion

Throughtout this paper, we revisit reward modeling for diffusion models from a diffusion-native perspective and show that latent diffusion backbones can serve as practical and competitive reward models for preference alignment. By formulating preference learning on noisy diffusion states with noise-calibrated uncertainty and enabling inference-time noise ensembling, our proposed DiNa-LRM substantially strengthens diffusion-based rewards and brings their preference prediction closer to strong VLM judges. More importantly, operating natively in latent space yields a structural advantage for post-training alignment: DiNa-LRM provides an optimization-friendly reward signal that improves proxy and held-out golden metrics in tandem, while cutting resource cost substantially (e.g., $51.4\%$ peak VRAM and large reductions in reward/optimization FLOPs). These results suggest that diffusion-native latent reward modeling is a promising alternative to pixel-space VLM rewards.

**Limitations and Future Work**  Our reward is learned and evaluated in the latent space of a specific diffusion backbone, which makes optimization efficient but does not guarantee cross-backbone transfer. Improving generality remains an open direction. Moreover, latent-space modeling may under-emphasize certain pixel-level artifacts that can be amplified during preference optimization. For example, we observe that long-horizon reward optimization can also lead to over-optimization or reward-hacking behaviors, such as spurious object insertion or stylistic drift, as discussed in subsection D.1. Future work includes (i) training on stronger and more unified backbones to improve generality, (ii) adding lightweight pixel-space regularization or perceptual constraints to penalize latent-invisible artifacts, and (iii) exploring generative or feedback-rich reward modeling that produces dense rewards.

# Acknowledgements

This work was supported in part by the National Natural Science Foundation of China (Grant No. 62372480), in part by 2025 Tencent AI Lab Rhino-Bird Focused Research Program, in part by HKUST-MetaX Joint Lab Fund (No. METAX24EG01-D), and in part by a grant from the NSFC/RGC Collaborative Research Scheme sponsored by the Research Grants Council of the Hong Kong Special Administrative Region, China and National Natural Science Foundation of China (Project No. CRS_HKUST605/25).

# Impact Statement

This paper develops a diffusion-native reward model for preference learning in text-to-image generation, facilitating automated evaluation and post-training optimization of diffusion models. However, as reward models guide generative behavior, biases inherent in preference datasets may be inherited or amplified during optimization, potentially yielding stereotypical or discriminatory outputs. These risks necessitate responsible deployment practices, including rigorous auditing during optimization, and the integration of robust safety protocols within generative pipelines.

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

# Appendix

Our Appendix consists of 5 sections. Readers can click on each section number to navigate to the corresponding section:

- Section A analyzes the uncertainty introduced by stochastic noising in diffusion-native reward evaluation, and examines whether such score-level randomness affects pairwise preference decisions.

- Section B provides detailed algorithmic procedures for DiNa-LRM, including noise-calibrated Thurstone training, inference-time noise ensembling, and the implementation of reward-gradient alignment with ReFL.

- Section C presents extensive experimental results, including comparisons with LRM-SDXL, additional ablation studies, generalization and transferability analyses across diffusion backbones, and broader alignment experiments with both ReFL and Flow-GRPO-Fast.

- Section D discusses practical issues and broader implications of diffusion-native reward modeling, including reward hacking behaviors under long-horizon optimization and potential extensions to long video generation, image/video editing, and audio-visual generation.

- Section E provides additional experimental details, including training settings, implementation choices, evaluation protocols, and other reproducibility-related configurations.

## A. Uncertainty Analysis

Our reward model is evaluated on noise-conditioned inputs. Each evaluation samples Gaussian noise to construct a perturbed state $x_t$. Consequently, the predicted score is stochastic, and the pairwise ranking decision for a fixed comparison pair may vary across repeated noise draws. In this section, we quantify this effect on a subset of **1,000** preference pairs randomly sampled from the HPDv3 test set.

For a fixed preference pair $(x_0^+, x_0^-)$ and a fixed evaluation noise level $t$, we repeat the noising-and-scoring process $K = 10$ times and obtain $K$ pairwise decisions:

$$d_k = \mathbb{I}\Big[r_\theta(x_{t,k}^+, t, c) > r_\theta(x_{t,k}^-, t, c)\Big], \quad k = 1, \ldots, K. \tag{14}$$

Let $p = \frac{1}{K}\sum_{k=1}^{K} d_k$ denote the empirical probability that the model ranks the chosen sample above the rejected one under stochastic noising. We measure decision instability using the **variation ratio**:

$$\text{VR}(t) = 1 - \max(p, 1 - p) = \min(p, 1 - p), \tag{15}$$

which ranges from 0 (fully consistent decisions) to 0.5 (maximally ambiguous decisions). We report the average $\text{VR}(t)$ over evaluation pairs for each noise level $t$.

To characterize score-space behavior, we compute the **pairwise margin** $\Delta r_k = r_k^+ - r_k^-$ and report its mean

$$\mu_{\Delta r}(t) = \frac{1}{K}\sum_{k=1}^{K} \Delta r_k, \tag{16}$$

which reflects how strongly the model separates the chosen sample from the rejected one on average. In addition, we quantify score sensitivity for a single sample by the **variance of its predicted score** across the $K$ runs,

$$\text{Var}(r) = \frac{1}{K-1}\sum_{k=1}^{K} \big(r_k - \bar{r}\big)^2, \quad \bar{r} = \frac{1}{K}\sum_{k=1}^{K} r_k, \tag{17}$$

and report $\text{Var}(r)$ averaged over all evaluated samples at each $t$.

*Table 4.* **Uncertainty Analysis.** We repeat the evaluation $K{=}10$ times per pair at each noise level $t$. VR measures decision instability, $\mu_{\Delta r}$ is the mean pairwise margin, and $\text{Var}(r)$ is the average per-sample score variance across repeated noise samples.

| Eval. $t$ | VR $\downarrow$ | $\mu_{\Delta r}$ $\uparrow$ | $\text{Var}(r)$ $\downarrow$ |
|---|---|---|---|
| 0.01 | 0.003 | 0.267 | $6.57 \times 10^{-4}$ |
| 0.2 | 0.015 | 0.564 | $4.53 \times 10^{-3}$ |
| 0.4 | 0.023 | 0.929 | $2.39 \times 10^{-2}$ |
| 0.6 | 0.029 | 1.252 | $1.18 \times 10^{-1}$ |
| 0.8 | 0.063 | 1.405 | $2.25 \times 10^{-1}$ |

Table 4 shows that decision stochasticity remains limited across noise levels: $\text{VR}(t)$ stays small even at high noise (e.g., 0.063 at $t{=}0.8$), indicating that pairwise decisions are typically consistent across different noise sampling. As expected, the score variance $\text{Var}(r)$ increases substantially with $t$, reflecting stronger randomness in highly noisy states. Overall, these results indicate that while stochastic noising introduces score-level uncertainty, **it induces only mild decision-level ambiguity**, and the obtained scores from a diffusion reward model are reliable.

## B. Detailed Algorithmic Procedures

In this section, we provide a formal description of the algorithmic frameworks for both the training and inference phases of DiNa-LRM, as well as the implementation of ReFL in subsection 5.4.

### B.1. Training with Noise-Calibrated Thurstone

As presented in algorithm 1, we detail the optimization for our latent reward model. The procedure emphasizes our noise-calibrated preference learning (subsection 4.1), where the comparison variance $\sigma_u^2(t)$ is explicitly modulated by the diffusion noise level $\sigma(t)$. Notably, the entire process operates within the VAE latent space, bypassing the need for image-space decoding and significantly improving training throughput.

---

**Algorithm 1:** DiNa-LRM Training Procedure

---

**Input** : Preference pairs $(\mathbf{x}_0^+, \mathbf{x}_0^-, \mathbf{c}, y)$, schedule $(\alpha_t, \sigma_t)$, $k, \sigma_u$
**Output** : Reward model $r_\theta$

1 **for** *each iteration* **do**
2     # Step 1: Sample diffusion noise level
3     $t \sim \mathcal{U}(0,1), \boldsymbol{\epsilon} \sim \mathcal{N}(0, \mathbf{I})$ ;                            // Uniform t sampling
4     # Step 2: Forward noising in latent space (Equation 1)
5     $\mathbf{x}_t^+ = \alpha(t)\mathbf{x}_0^+ + \sigma(t)\boldsymbol{\epsilon}$
6     $\mathbf{x}_t^- = \alpha(t)\mathbf{x}_0^- + \sigma(t)\boldsymbol{\epsilon}$
7     # Step 3: Noise-calibrated reward inference
8     $r^+ = \texttt{r\_theta}(\mathbf{x}_t^+, t, \mathbf{c})$
9     $r^- = \texttt{r\_theta}(\mathbf{x}_t^-, t, \mathbf{c})$
10     $\sigma_u^2(t) = k\sigma^2(t) + \sigma_u^2$ ;                      // Equation 7: Calibrated uncertainty
11     # Step 4: Probabilistic preference modeling (Equation 8)
12     $\Delta r = r^+ - r^-$
13     $\hat{p}_\theta = \Phi\left(\Delta r / \sqrt{2\sigma_u^2(t)}\right)$
14     # Step 5: Fidelity loss optimization (Equation 9)
15     $\mathcal{L}_{\text{fid}} = 1 - \sqrt{y\hat{p}_\theta + (1-y)(1-\hat{p}_\theta)}$
16     $\theta \leftarrow \theta - \eta\nabla_\theta\mathcal{L}_{\text{fid}}$

---

### B.2. Inference-Time Noise Ensembling

algorithm 2 illustrates our strategy for inference-time scaling (subsection 4.3). Unlike standard reward models that evaluate a single point estimate, DiNa-LRM allows for the aggregation of multiple noise-conditioned views of the same clean sample. By concatenating FiLM-modulated tokens across $K$ distinct timesteps, the query-based head can attend to a global context of the diffusion trajectory. This token-level ensembling serves as a test-time scaling skill, where increased inference compute can be traded for improved reward stability and alignment accuracy.

### B.3. Implementation of Reward-Gradient Alignment (ReFL)

To evaluate the practical effectiveness of DiNa-LRM as a differentiable optimization signal, we integrate it into the ReFL framework for post-training alignment. Given a text prompt $c \sim \mathcal{D}_{prompts}$, we initiate the denoising process from a random noise latent $\boldsymbol{x}_T \sim \mathcal{N}(0, \mathbf{I})$. We perform the initial $N-1$ denoising steps without tracking gradients. This produces an intermediate latent $\boldsymbol{x}_{t_1}$ that captures the basic semantic structure of the image. In the final denoising step(s), we enable gradient tracking through the diffusion UNet or Transformer backbone. We compute the one-step-predicted clean latent $\hat{\boldsymbol{x}}_0$. Then we adopt DiNa-LRM to produce a scalar reward $r_\phi(\hat{\boldsymbol{x}}_0, \boldsymbol{c}) = r_\phi\left(\alpha(t^*)\hat{\boldsymbol{x}}_0 + \sigma(t^*)\boldsymbol{\epsilon}, t^*, c\right)$, where $t^* = 0.4$ in our experiment setting. Then we calculate the ReFL loss and backpropagate it:

$$\mathcal{L}_{ReFL} = -\mathbb{E}_{\hat{\boldsymbol{x}}_0}[r_\phi(\hat{\boldsymbol{x}}_0, \boldsymbol{c})] \tag{18}$$

---

**Algorithm 2:** Inference-Time Scaling via Noise Ensembling

---

**Input** : Clean sample $\mathbf{x}_0$, prompt $\mathbf{c}$, evaluation timesteps $\{t_k\}_{k=1}^K$
**Output** : Aggregated reward score $\hat{r}$

---

1 # Step 1: Multi-timestep feature extraction
2 $\mathcal{V}, \mathcal{T} \leftarrow [\,], [\,]$ ;                                                              // Initialize token storage
3 **for** $k = 1$ **to** $K$ **do**
4      # Perturb sample to the specified noise level
5      $\mathbf{x}_{t_k} = \alpha(t_k)\mathbf{x}_0 + \sigma(t_k)\boldsymbol{\epsilon}$
6      # Extract features from diffusion backbone (Equation 10)
7      $\mathbf{F}_{vis}, \mathbf{F}_{txt} = \texttt{Backbone}(\mathbf{x}_{t_k}, t_k, \mathbf{c})$ ;                    // Backbone feature exaction
8      # Step 2: Timestep-conditioned adaptation
9      $\mathbf{V}_{t_k} = \texttt{FiLM\_Modulate}(\mathbf{F}_{vis}, t_k)$ ;                         // Feature Modulation
10      $\mathbf{T}_{t_k} = \texttt{FiLM\_Modulate}(\mathbf{F}_{txt}, t_k)$
11      Append $\mathbf{V}_{t_k}$ to $\mathcal{V}$, $\mathbf{T}_{t_k}$ to $\mathcal{T}$

12 # Step 3: Token-level ensembling (Equation 13)
13 $\mathbf{V}_{\text{ensemble}} = \texttt{Concat}(\mathcal{V})$ ;                                    // Join tokens across timesteps
14 $\mathbf{T}_{\text{ensemble}} = \texttt{Concat}(\mathcal{T})$
15 # Step 4: Global query-based scoring
16 $\hat{r} = \texttt{Q\_Former}(\mathbf{V}_{ens}, \mathbf{T}_{ens})$
17 **return** $\hat{r}$

---

## C. Extensive Experiments

### C.1. Comparison with LRM-SDXL in alignment

In the main paper, our alignment experiments are conducted on the SD3.5-M backbone, whereas LRM-SDXL (Zhang et al., 2025b) is built upon the SDXL backbone. To provide a fairer comparison with this diffusion-based reward baseline, we additionally instantiate DiNa-LRM on SDXL and compare it with LRM-SDXL from two perspectives: offline reward-model evaluation and ReFL-based alignment.

We first compare the pairwise preference accuracy of LRM-SDXL and DiNa-LRM instantiated on the same SDXL backbone. As shown in Table 5, DiNa-LRM substantially outperforms LRM-SDXL across most benchmarks, improving the average accuracy. This suggests that the proposed diffusion-native latent preference formulation provides a stronger reward-learning objective even when using the same SDXL family backbone.

*Table 5.* **Pairwise Preference Accuracy Comparison with LRM-SDXL.** We compare LRM-SDXL and DiNa-LRM instantiated on the SDXL backbone. The better result is shown in **bold**.

| Method | Pairwise Preference Accuracy (%) | | | | |
|---|---|---|---|---|---|
| | ImageReward | HPDv2 | HPDv3 | GenAI Bench | Avg |
| LRM-SDXL | 60.35 | 71.19 | 53.80 | 61.58 | 61.73 |
| DiNa-LRM (SDXL) | **60.86** | **81.03** | **74.87** | **69.44** | **71.55** |

We further evaluate both reward models in the same ReFL alignment setting on SDXL. Following the protocol in the main paper, we treat the optimized reward as the proxy score and report PickScore as an external held-out golden metric, which is not used during optimization. All configurations are kept identical across the two runs, including the generator backbone, prompts, optimization steps, and ReFL implementation; only the reward model is changed.

As shown in Figure 5, DiNa-LRM also leads to stronger alignment dynamics than LRM-SDXL under the same SDXL-based ReFL setting. Together with the offline reward-model comparison, these results validate DiNa-LRM as an effective and optimization-friendly reward model for diffusion-model alignment.

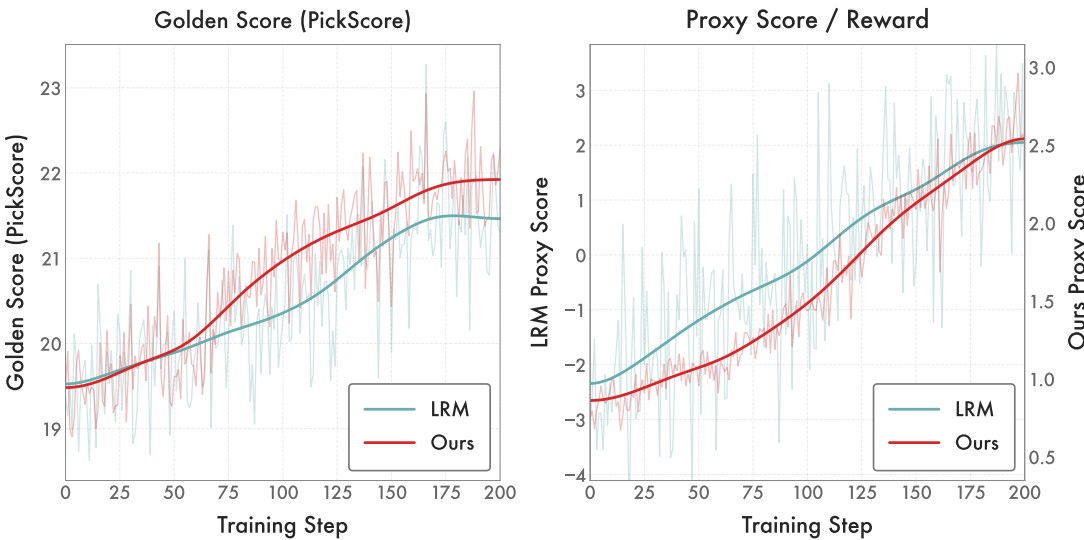

*Figure 5.* **Training Curves (ReFL on SDXL).** We optimize SDXL with either LRM-SDXL or DiNa-LRM  (SDXL) as the proxy reward. We report the optimized proxy score *(right)* and an external held-out golden metric *(PickScore; left)*. DiNa-LRM  provides a stronger alignment signal, improving the held-out PickScore more effectively while the proxy reward increases consistently.

### C.2. Extensive Ablation Studies

**Impact of Reward Head Design**   Our reward head needs to aggregate multi-layer visual features together with textual features. This setting benefits from selective token-level fusion rather than fixed feature pooling, especially because our inference-time multi-noise ensembling concatenates features from multiple timesteps into a longer token context. To validate this design choice, we compare our gated Q-Former head with two simpler alternatives: **(i) Average Pooling**, which directly pools the aggregated features into a global representation, and **(ii) Transformer + Average Pooling**, which first applies a lightweight transformer over the fused tokens and then performs average pooling. As shown in Table Table 6, all three variants are reasonably effective, indicating that the diffusion-native preference formulation is not overly dependent on a specific reward-head architecture. Nevertheless, the gated Q-Former achieves the best average performance.

It is worth noting that our research prioritizes the design and verification of the *diffusion-native formulation* rather than exhaustive architectural tuning. While specific hyperparameters of the reward head may exert moderate influences on final scores, such optimal configurations often vary across different backbone architectures. Our preliminary investigations reveal that the inclusion of text features and the utilization of multi-level backbone features are critical factors for performance, whereas other architectural choices yield less significant impact.

*Table 6.* **Ablation on Reward Head Design.**

| Method | Pairwise Preference Accuracy (%) | | | | |
|---|---|---|---|---|---|
| | ImageReward | HPDv2 | HPDv3 | GenAI Bench | Avg |
| Average Pooling | 60.68 | 80.70 | 74.43 | 68.31 | 71.03 |
| Transformer + Average Pooling | **60.98** | 80.71 | 74.70 | 68.18 | 71.14 |
| Gated Q-Former (**Ours**) | 60.34 | **82.13** | **75.04** | **68.43** | **71.49** |

**Impact of Layer Depth**   Our default configuration utilizes features from 12 layers (specifically indices 4, 8, 12) extracted from the diffusion backbone. To evaluate the sensitivity of the reward model to feature granularity, we compare configurations using 8, 12, 16 and 20 layers. As shown in Table Table 7, we observe a consistent and positive correlation between the number of feature layers and pairwise preference accuracy across all benchmarks, suggesting that a sufficient hierarchical representation is necessary to capture complex human preferences. Due to the limited computational resources, we conduct our primary ablation experiments using the 12-layer configuration as a representative setting.

*Table 7.* **Ablation on Layer Depth.**

| Configure | Pairwise Preference Accuracy (%) | | | | |
|---|---|---|---|---|---|
| | ImageReward | HPDv2 | HPDv3 | GenAI Bench | Avg |
| 8 layers | 58.83 | 74.39 | 72.28 | 66.79 | 68.07 |
| 12 layers (**Ours**) | 60.34 | 82.13 | 75.04 | 68.43 | 71.49 |
| 16 layers | 61.13 | 82.57 | 75.16 | 68.52 | 71.85 |
| 20 layers | 61.94 | 83.44 | 75.37 | 70.29 | 72.76 |

### C.3. Generalization and Transferability

We further investigate the generality of the proposed diffusion-native reward formulation. Here, "general-purpose" does not mean that a single reward model trained on one backbone should directly transfer to arbitrary generators with different latent spaces or VAEs. Instead, our intended claim is that DiNa-LRM provides a reusable reward-model formulation for preference alignment, rather than an algorithm-specific step-level reward tied to a particular optimization procedure. To support this claim, we provide two complementary evaluations: (i) applying the same formulation to diverse diffusion backbones, and (ii) reusing a trained reward model to align another generator instance within the same latent space.

**Generalization Across Diverse Backbones.** To evaluate whether the proposed formulation is specific to SD3.5-M, we instantiate DiNa-LRM on four representative text-to-image backbones: SDXL (Podell et al., 2024), SD3.5-M (Esser et al., 2024), FLUX.1-Dev (Labs, 2024), and Z-Image-Turbo (Cai et al., 2025). These models cover three common architectural families: U-Net, dual-stream DiT, and single-stream DiT. For FLUX.1-Dev, we adopt features from the first 19 layers; for Z-Image-Turbo, we use the first 16 layers. As shown in Table 8, DiNa-LRM maintains strong performance across these different backbone families and consistently outperforms prior diffusion-based reward baselines by a large margin.

*Table 8.* **Generalization Across Different Diffusion Backbones.**

| Backbone | Pairwise Preference Accuracy (%) | | | | |
|---|---|---|---|---|---|
| | ImageReward | HPDv2 | HPDv3 | GenAI Bench | Avg |
| SDXL | 60.86 | 81.03 | 74.87 | 69.44 | 71.55 |
| SD3.5-M (12 layers) | 60.34 | 82.13 | 75.04 | 68.43 | 71.49 |
| FLUX.1-Dev (19 layers) | 59.03 | 81.21 | 72.57 | 66.67 | 69.87 |
| Z-Image-Turbo (16 layers) | 60.13 | 81.75 | 71.58 | 67.21 | 70.17 |

**Transfer within a shared latent space.** We further study whether a trained diffusion-native reward model can be reused to optimize another generator instance within the same latent space. Specifically, we train the reward model on SD3.5-M and use it to align SD3.5-L under the same ReFL setting. Since SD3.5-M and SD3.5-L share a compatible latent space, this experiment evaluates transferability across generator instances without introducing an additional VAE-space mismatch.

As shown in Figure 6, the held-out PickScore exhibits a clear upward trend throughout optimization, while the optimized proxy reward also increases steadily. This indicates that a reward model trained on SD3.5-M can provide an effective optimization signal for SD3.5-L when the two models operate in the same latent space. Importantly, the improvement is reflected not only in the optimized proxy reward but also in the held-out golden metric, suggesting that the transferred reward remains practically useful for alignment rather than merely fitting the proxy objective. This result does not imply unrestricted zero-shot transfer across arbitrary backbones with incompatible VAEs. Rather, it supports the practical transferability of diffusion-native rewards across generator instances that share a latent representation space. Together with the cross-backbone evaluation in Table 8, these results show that DiNa-LRM is a general reward-modeling formulation for diffusion-model preference alignment, rather than a method tied to one specific backbone or one specific alignment algorithm.

**Discussion about Scaling Behaviors** Interestingly, unlike the significant scaling behaviors observed in VLM-based reward models (Wu et al., 2025b), we do not observe a corresponding performance leap in benchmarks when scaling the underlying diffusion backbone from SD3.5-M (2B) (Esser et al., 2024) to Z-Image-Turbo (7B) (Cai et al., 2025) and FLUX.1-Dev (12B) (Labs, 2024). A more conservative yet practical conclusion is that our formulation remains effective across diverse

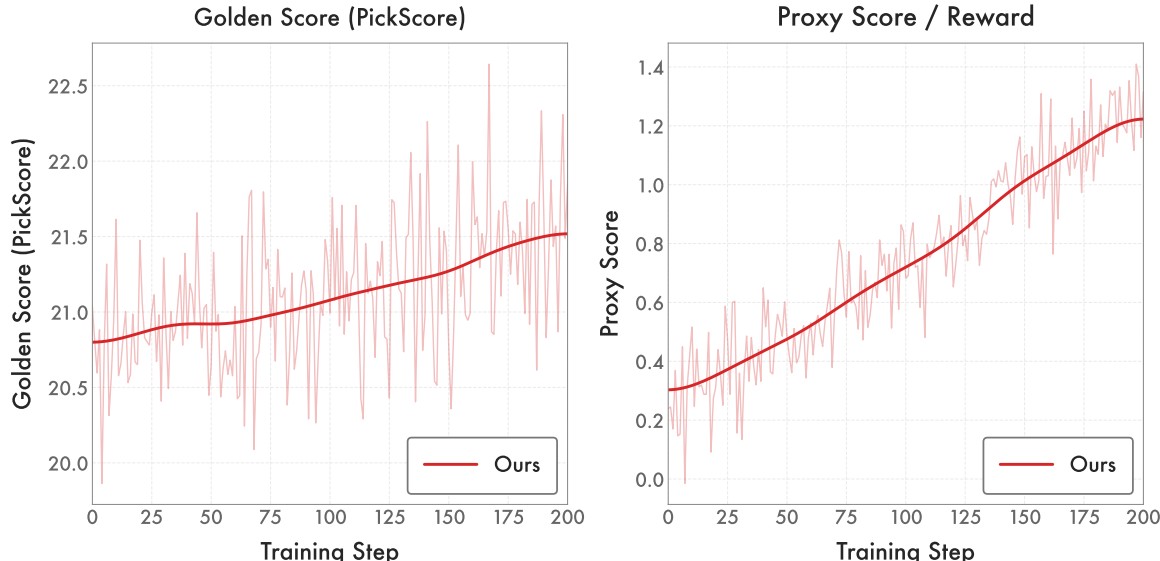

*Figure 6.* **Transfer from an SD3.5-M Reward Model to SD3.5-L Alignment.** We reuse a reward model trained on SD3.5-M to optimize SD3.5-L under ReFL. We report the held-out golden metric *(PickScore; left)* and the optimized proxy reward *(right)* across training steps.

backbones, providing similar performance regardless of the model scale. We hypothesize that *larger generative backbones often distribute their discriminative priors across a broader or deeper set of layers*, necessitating more intensive layer-wise search and feature aggregation to unlock their full reward potential. Due to limited computational resources, we did not perform exhaustive hyperparameter tuning or deeper layer searches for the 7B and 12B models.

Additionally, it appears that the performance of diffusion-native rewards may be more heavily bounded by the quality of the preference training data rather than the generation capacity of the backbone. Consequently, a promising direction for future work would be distilling high-performance pixel-space VLM rewards into these latent-space backbones to achieve both superior accuracy and efficient on-policy alignment.

### C.4. Alignment Experiments with Flow-GRPO

**Flow-GRPO Optimization** While ReFL serves as a gradient-based supervised alignment baseline, we further validate DiNa-LRM in an online reinforcement learning setting using Flow-GRPO (Liu et al., 2025a). To ensure rapid verification, we adopt **Flow-GRPO-Fast** variant, which utilizes a hybrid sampling strategy: only the first 3 SDE denoising steps are optimized with gradient tracking, while the remaining steps follow standard ODE sampling. Following the evaluation protocol in subsection 5.4, we designate PickScore as a held-out "golden metric" that does not participate in training, serving as an external indicator for monitoring over-optimization. Our RL configuration employs a group size of 18, with 32 prompts processed per optimization step. The training dynamics visualized in Figure 7 and intermediate sample visualizations visualized in Figure 8 demonstrate that DiNa-LRM provides a stable and effective reward signal for complex online RL trajectories.

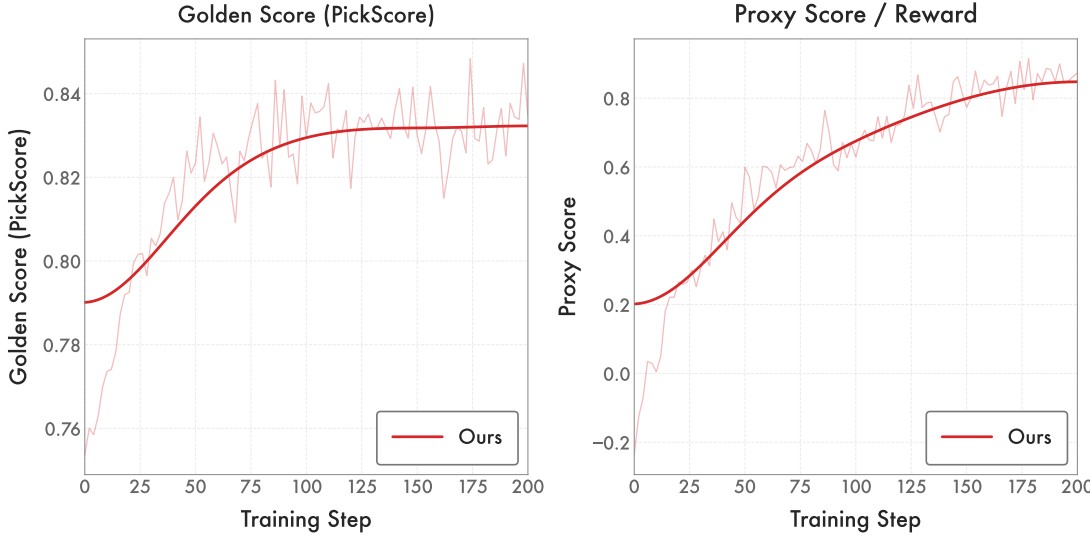

*Figure 7.* **Training Curves (Flow-GRPO-Fast on SD3.5-M)**. We optimize with DiNa-LRM as the **proxy** reward. We report the optimized proxy score *(right)* and an external held-out **golden metric** *(PickScore; left)*.

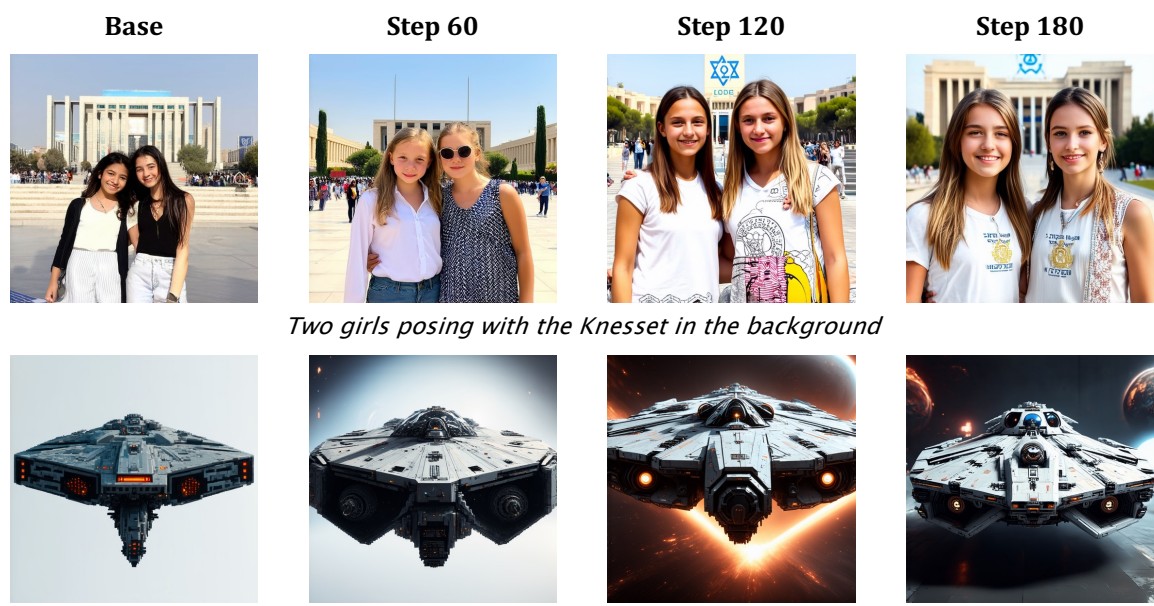

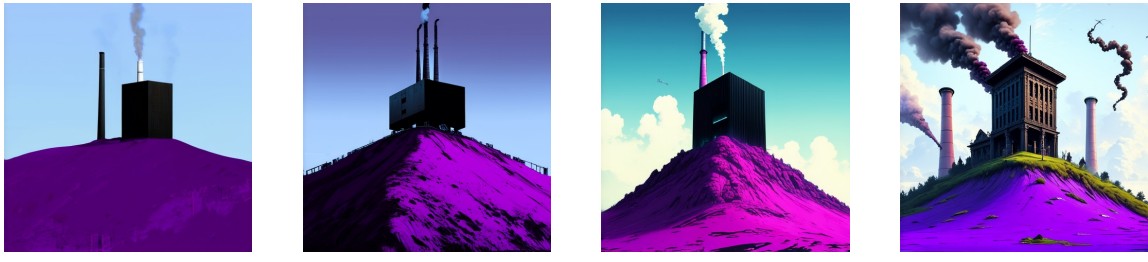

*Figure 8.* Qualitative Evolution during Flow-GRPO-Fast Alignment.

# D. Extensive Discussions

## D.1. Discussion about Reward Hacking

In the context of preference alignment, reward hacking remains a critical challenge where the generative model exploits the reward function to achieve high scores without genuine quality improvements. Similar to observations in prior works (Liu et al., 2025a), we find that during the early stages of alignment, the proxy reward and the held-out golden metric (PickScore) increase in tandem, indicating effective preference learning. However, as optimization progresses into the long-horizon regime, the growth of the golden metric stagnates or even declines, while the proxy reward begins to fluctuate significantly or reaches a plateau. During our experiments, we identify two distinct hacking patterns (see Figure 9):

- **Spurious Human Injection.** We notice a tendency where the reward model assigns higher scores to images featuring human subjects. Consequently, the generative model occasionally incorporates human figures into scenes even when they are not explicitly requested by the prompt.

- **Stylistic Drift toward Animation.** The model exhibits a strong tendency to shift toward an anime or illustrative style when the prompt does not specify realistic or photographic outputs.

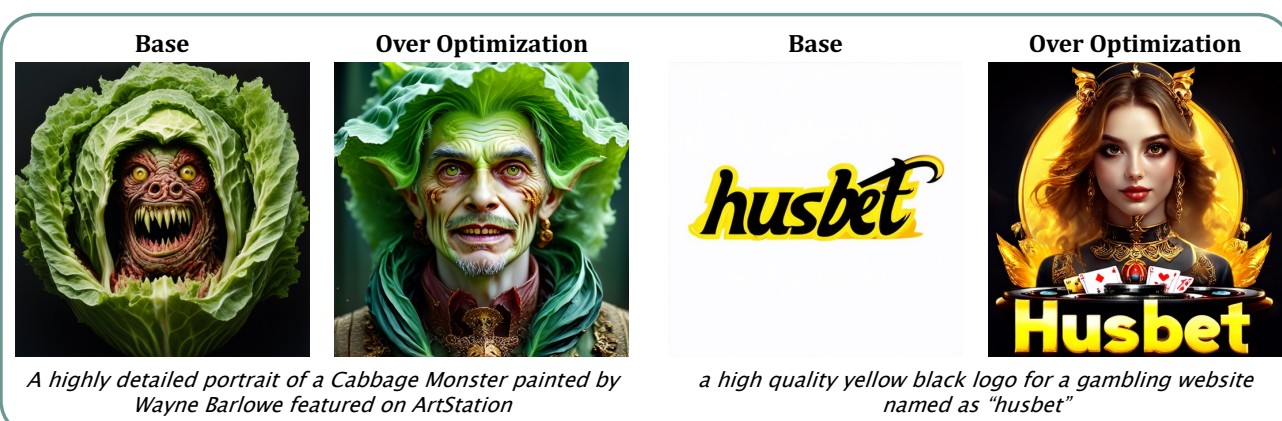

*A highly detailed portrait of a Cabbage Monster painted by Wayne Barlowe featured on ArtStation*

*a high quality yellow black logo for a gambling website named as "husbet"*

**Hacking Pattern 1: Spurious Human Injection**

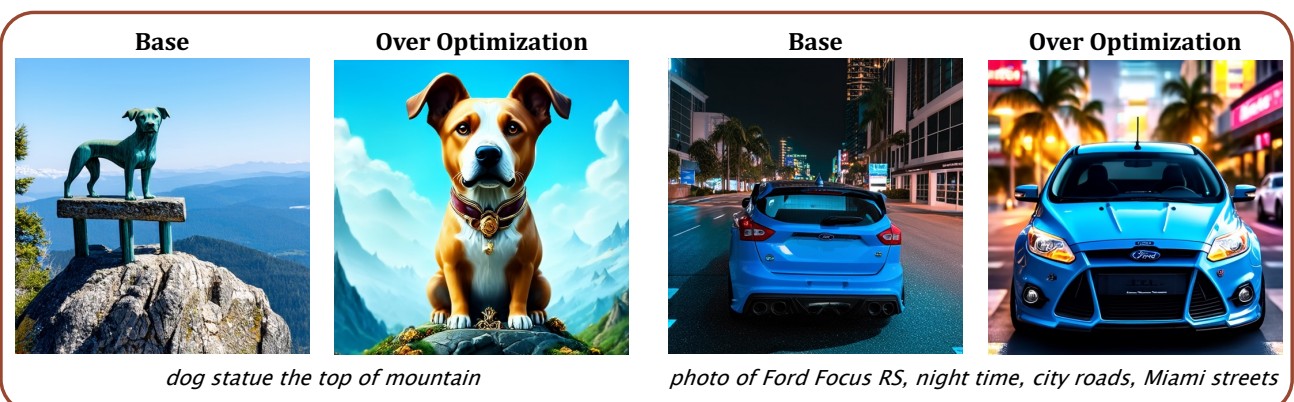

*dog statue the top of mountain*

*photo of Ford Focus RS, night time, city roads, Miami streets*

**Hacking Pattern 2: Stylistic Drift toward Animation**

*Figure 9.* **Visualization of Reward Hacking Patterns.** We present representative failure cases observed during the late-stage optimization (step 780) of the Flow-GRPO-Fast pipeline. These patterns illustrate two primary forms of over-optimization: **(1). Spurious Human Injection. (2). Stylistic Drift toward Animation.**

To mitigate these issues, regularization techniques are not only beneficial but also essential. We demonstrate that the pretraining loss in ReFL (Xu et al., 2023), along with KL-divergence constraints and regulated clipping ratios in Flow-GRPO (Liu et al., 2025a), effectively stabilizes the optimization trajectory and delays the onset of hacking.

Additionally, we also notice a prioritization of *Aesthetic Quality over Semantic Fidelity*. The reward model tends to favor high-quality, visually appealing images even if they exhibit minor prompt-mismatch, over lower-quality images that strictly follow the prompt. While the final rewards remain positively correlated with prompt alignment, this vision-centric bias is likely a shared limitation among most vision-foundation-based reward models. In practice, a more robust alignment strategy involves combining DiNa-LRM with metrics specialized in text-image alignment to provide a more balanced and comprehensive feedback signal.

### D.2. Broader Application Scenarios

Although this work focuses on text-to-image preference modeling and alignment, diffusion-native reward modeling may also serve as a useful building block for broader visual generation scenarios. Recent generative models have moved beyond single-image synthesis toward more complex settings, including long video generation (Lu et al., 2025b; Zhang et al., 2026), image/video editing (Wei et al., 2025; Ye et al., 2026a), joint audio-visual generation (Kong et al., 2025; Zhong et al., 2025), and other AIGC applications (Liu et al., 2026; Xue et al., 2025a). These scenarios require reward models to evaluate multiple intertwined factors, such as semantic alignment, temporal consistency, identity or content preservation, physical plausibility, audio-visual synchronization, and instruction following. These further motivate reward models that are discriminatively reliable, efficient to query during optimization, and compatible with the latent generative process. Extending DiNa-LRM to these broader scenarios would require task-specific preference data and evaluation protocols, but the diffusion-native latent formulation provides a practical starting point for efficient reward evaluation and post-training alignment.

## E. Hyperparameters Details

Here, we provide all the detailed hyperparameters for: (i) Reward Modeling Training (subsection 5.2); and (ii) Text-to-Image Alignment (subsection 5.4).

*Table 9.* Hyperparameters for Reward Modeling

| Training | |
|---|---|
| Training strategy | LoRA |
| LoRA alpha | 128 |
| LoRA dropout | 0.0 |
| LoRA R | 64 |
| LoRA target-modules | q_proj,k_proj,v_proj,o_proj |
| Optimizer | AdamW |
| Learning rate | 5e-5 |
| EMA Decay | 0.995 |
| Epochs | 1 |
| Batch size | 64 (after accumulation) |
| **Model Architecture on SD-3.5-M** | |
| Number of Transformer Blocks | 12 |
| Visual Feature Layers Index | 4, 8, 12 |
| Textual Feature Layers Index | 4, 8, 12 |
| Query Number | 4 |
| **Timestep Sampling** | |
| Sampling Strategy | Uniform Sample ($t \sim \mathcal{U}(0, 1)$) |

*Table 10.* Hyperparameters for Preference Alignment (ReFL) on SD3.5-M

| Training | |
|---|---|
| Training strategy | LoRA |
| LoRA alpha | 64 |
| LoRA dropout | 0.0 |
| LoRA R | 32 |
| LoRA target-modules | q_proj,k_proj,v_proj,o_proj |
| Optimizer | Adam |
| Learning rate | 3e-5 |
| Training Steps | 150 |
| Batch size | 256 (after accumulation) |
| **Algorithm of ReFL** | |
| Rollout Sample Steps | 40 |
| Stop Grad Step | [30, 39] |

