# OpenReview forum: "Beyond VLM-Based Rewards: Diffusion-Native Latent Reward Modeling"
_ICML.cc/2026/Conference — ICML 2026 regular_

### Official Review · Reviewer_AC5i · 2026-03-05

**Soundness:** 3
**Presentation:** 3
**Significance:** 3
**Originality:** 3
**Overall Recommendation:** 5
**Confidence:** 3

**Summary:**

This paper revisits reward modeling for diffusion-based image generation and proposes a diffusion-native alternative to pixel-space VLM rewards. The key idea is to formulate preference learning directly in the latent diffusion space, making the reward model better aligned with the underlying generative backbone and more efficient during optimization. The proposed approach is built on a pretrained diffusion model and incorporates noise-aware preference modeling and inference-time ensembling. Empirically, it substantially improves over prior diffusion-based reward baselines, approaches the performance of strong VLM rewards, and demonstrates clear efficiency and optimization advantages in alignment experiments.

**Compliance With Llm Reviewing Policy:**

Affirmed.

**Final Justification:**

The rebuttal confirmed my initial assessment of the paper.

**Key Questions For Authors:**

- Empirically, the reward accuracy peaks at intermediate noise levels (e.g., t∈[0.3,0.7]), whereas I would intuitively expect evaluation at t=0 (clean latent) to be most semantically faithful. Could the authors provide deeper intuition for why intermediate noisy states are more discriminative for preference prediction? Is this effect specific to diffusion backbones, or does it reflect a more general representation property?
- Since the reward operates entirely in the VAE latent space, how sensitive is the approach to the choice of latent autoencoder? In particular, could latent-space reward modeling overlook certain pixel-level artifacts that may become amplified during long-horizon alignment?

**Limitations:**

yes

**Strengths And Weaknesses:**

- Soundness: The method is technically well grounded and internally consistent. The diffusion-native preference formulation is coherent with the training dynamics of diffusion models, and the noise-calibrated Thurstone likelihood is a reasonable extension of standard preference modeling. The experimental evaluation is thorough: the authors compare against strong CLIP and VLM based reward models, include meaningful ablations (timestep schedules, variance modeling, ensembling), and validate the approach in an actual alignment setting. The reported efficiency gains (VRAM and FLOPs reductions) further strengthen the practical credibility of the method. Overall, the claims are well supported by empirical evidence.
- Presentation: The paper is generally clear and well structured. The high-level motivation of reducing pixel/latent mismatch and improving efficiency is easy to follow. The method section logically develops from the diffusion-native formulation to the architecture and inference-time scaling. The ablations are well organized and help clarify the impact of design choices.
- Significance: Overall, the authors analyze the important topic of designing reward models that are both effective and optimization-friendly for large diffusion systems. Given the growing reliance on reward-based post-training for generative models, improving efficiency and compatibility with latent diffusion backbones is highly relevant. While the gains in raw reward accuracy are moderate, the demonstrated improvements in optimization dynamics and computational cost make the contribution practically meaningful.
- Originality: The idea of using diffusion models for discriminative tasks is not entirely new, but framing reward modeling in a fully diffusion-native, latent-space manner is a meaningful conceptual step. The integration of noise-aware preference modeling with timestep-aware scoring and inference-time noise ensembling results in a cohesive and well-motivated design.

---

> ### Author Rebuttal · Authors · 2026-03-31
>
> We thank the reviewer for the positive assessment of our paper and for the insightful questions regarding the role of intermediate noise levels and the sensitivity of latent-space reward modeling. We answer these questions below.
>
>
> ---
>
> `[Q1] Why are intermediate noisy states more discriminative than clean latents?`
>
> Thank you for this important question. Our intuition is that **intermediate noise levels provide a better trade-off between semantic fidelity and discriminative abstraction**. At $t=0$, the clean latent preserves more low-level detail and nuisance variation, which is not always most relevant for preference comparison. At intermediate noise levels, part of this low-level variation is suppressed while the main semantic and compositional information is still preserved, making preference differences easier to separate. When the noise becomes too large, semantic information is eventually degraded, so the reward accuracy decreases again. This leads to the observed “intermediate optimum.”
>
> We believe this effect is driven in part by **the diffusion-specific representation property**: the backbone is pretrained to denoise **"noisy"** states, so its internal features are naturally well structured on nonzero-noise inputs rather than only at the clean endpoint. We will clarify this intuition in the revision.
>
> ---
>
> `[Q2-1] Sensitivity to the latent autoencoder.`
>
> This is also an important point. Since our reward operates entirely in the latent space defined by the underlying autoencoder, its behavior is tied to that representation. In general, changing the autoencoder alters the latent space, so transfer across models with different latent spaces is not guaranteed and may require retraining or adaptation.
>
>
> While transfer across different latent spaces is not guaranteed, a key practical question is **whether the reward can generalize across different generators that share the same latent space**. To probe this, we train the reward model on **SD3.5-Medium** and use it to align **SD3.5-Large** under the same ReFL setup. We report both a held-out golden metric (PickScore) and the optimized proxy reward across training steps.
>
> **Transfer from SD3.5-Medium reward model to SD3.5-Large alignment.**
>
> $$
> \begin{array}{lccccc}
> \hline
> \text{Metric} & \text{Step 0} & 50 & 100 & 150 & 200 \newline
> \hline
> \text{Held-out PickScore} & 20.80 & 20.92 & 21.08 & 21.27 & 21.52 \newline
> \text{Proxy Reward (DiNa-LRM)} & 0.24 & 0.51 & 0.73 & 1.09 & 1.29 \newline
> \hline
> \end{array}
> $$
>
> These results show that the reward learned on SD3.5-Medium can be successfully reused to optimize SD3.5-Large, providing additional evidence that the method is **not tied to a single generator instance** when the latent space is shared.
>
> ---
>
> `[Q2-2] Possible pixel-level artifacts.`
>
> We agree that latent-space reward modeling may under-emphasize certain pixel-level artifacts that are weakly represented in latent space and only become clearly visible after decoding.
> As discussed in the Limitations section(*Section 6*), *"latent-space modeling may underemphasize certain pixel-level artifacts that can be amplified during preference optimization."* In our long-horizon optimization experiments, we do observe signs of such hacking behavior.
> We consider this a plausible limitation of latent-space reward modeling. A natural mitigation is to combine the latent-space reward with complementary pixel-space constraints, such as lightweight pixel-space regularization losses or hybrid reward formulations. We will make this discussion clearer in the revision.

---

> > ### Author Rebuttal · Reviewer_AC5i · 2026-04-02
> >
> > I thank the authors for answering my questions. After reading the answers, seeing the added results, and also seeing the additional ablations provided to the other reviewers, I decided to keep my score.

---

> > > ### Author Response · Authors · 2026-04-02
> > >
> > > Thank you for your acknowledgement and for your continued support. We sincerely appreciate the time and effort you took to read our responses provided during the rebuttal. We are glad that these clarifications helped further demonstrate the value of our work. We are grateful for your positive assessment and continued support of our work.

---

### Official Review · Reviewer_QdW1 · 2026-03-08

**Soundness:** 2
**Presentation:** 2
**Significance:** 2
**Originality:** 2
**Overall Recommendation:** 2
**Confidence:** 4

**Summary:**

This paper proposes DiNa-LRM, a reward model for preference alignment of diffusion-based image generators that operates natively in the VAE latent space. The key ideas are: (1) a noise-calibrated Thurstone preference formulation where comparison uncertainty scales with the diffusion noise level, (2) a timestep-conditioned reward head built on a pretrained SD3.5-Medium backbone using FiLM modulation and a gated Q-Former, and (3) inference-time noise ensembling that aggregates features across multiple timesteps for more robust scoring. The method substantially outperforms prior diffusion-based reward models and approaches VLM-based rewards in pairwise preference accuracy, while offering significant computational savings (51.4% VRAM reduction, 71.1% reward FLOPs reduction) during preference optimization.

**Compliance With Llm Reviewing Policy:**

Affirmed.

**Final Justification:**

We thank the authors for their effort in preparing the rebuttal. However, after careful consideration, the responses do not fully resolve my concerns, and I maintain my score.

**Key Questions For Authors:**

Please see weakness.

**Limitations:**

yes

**Strengths And Weaknesses:**

**Strengths**

1. **Substantial efficiency gains.** By avoiding decoding into pixel space during ReFL optimization, the reductions in memory and FLOPs are substantial, offering practical value for real-world deployment.
2. **Comprehensive ablations.** The paper validates design choices, including timestep schedules, the presence or absence of noise calibration, backbone adaptation strategies, and inference-time noise levels, making each component's contribution easy to understand.

**Weaknesses**

1. **Limited novelty.** Using the diffusion model backbone as a reward model is not particularly novel[1]. The noise-calibrated variance extension is natural, but it has also been discussed in prior work[2]. While the overall system integration is sensible, the conceptual leap beyond prior work is modest.
2. **Lack of cross-backbone transferability.** As the authors themselves acknowledge, the method is specifically designed and evaluated on SD3.5-Medium only. No experiments demonstrate whether DiNa-LRM transfers to diffusion models with entirely different VAEs, such as Flux or Qwen-Image. Even transferring to other models that share the same VAE remains untested. The rebuttal also only provides results on SD3.5, which does not resolve this concern. For a method that aims to serve as a general-purpose reward model, this is a significant limitation that undermines the claims' generality.
3. **Narrow VLM comparison.** The VLM comparison is centered primarily on HPSv3 (Qwen2-VL-7B), which is a regression-head reward model. However, the title "Beyond VLM-Based Rewards" sets a high bar that the experiments do not fully meet. Recent work in image and video generation evaluation has increasingly adopted large-scale VLMs such as GPT-4o, Gemini, and Qwen3-VL as preference judges [3,4,5,6], and these VLM-as-a-judge approaches represent a significant and growing branch of VLM-based reward modeling. Without any comparison or discussion of this paradigm, the scope of the paper's central claim remains limited.

[1] Video generation models are good latent reward models. arXiv2025.
[2] A general framework for inference-time scaling and steering of diffusion models. ICML2025.
[3] Viescore: Towards explainable metrics for conditional image synthesis evaluation. ACL2024.
[4] Boosting text-to-video generative model with MLLMs feedback. NeurIPS2024.
[5] Boost your own human image generation model via direct preference optimization with ai feedback. CVPR2025.
[6] Inference-time text-to-video alignment with diffusion latent beam search. NeurIPS2025.

---

> ### Author Rebuttal · Authors · 2026-03-31
>
> We thank the reviewer for the careful reading and for recognizing the substantial efficiency gains and the comprehensive ablations in our paper. We address the main concerns below.
>
> ---
>
> `[W1] Limited novelty.`
>
> We thank the reviewer for raising this concern.
> We respectfully disagree that the contribution is limited to simply reusing a diffusion backbone as a reward model or to a straightforward variance extension.
> We do not claim that the broad idea of using diffusion backbones for reward modeling is entirely new. In contrast, the paper already discusses prior diffusion-based reward modeling in *Section 2*.
>
> Our contribution is to study general-purpose preference reward modeling directly in diffusion latent space. We introduce a coherent formulation consisting of **(i) a noise-calibrated Thurstone likelihood** for noisy latent states, **(ii) an effective latent reward architecture**, and **(iii) inference-time multi-noise ensembling**. We also provide systematic evidence that this formulation substantially improves over prior diffusion-based reward baselines while enabling cheaper preference optimization. We will make this positioning clearer in the revision.
>
> ---
>
> `[W2-1] Lack of cross-backbone transferability.`
>
> Thank you for highlighting this concern. To address this, we additionally evaluate DiNa-LRM on **four backbones: SDXL, SD3.5-Medium, FLUX.1-Dev, and Z-Image-Turbo**, covering three common text-to-image-model families: **U-Net, dual-stream DiT, and single-stream DiT.**
>
> **Cross-backbone reward-model evaluation (pairwise accuracy, %).**
>
> $$
> \begin{array}{llccccc}
> \hline
> \text{Method} & \text{Backbone} & \text{ImageReward} & \text{HPDv2} & \text{HPDv3} & \text{GenAI Bench} & \text{Avg} \newline
> \hline
> \text{LRM-SD1.5} & \text{SD1.5 (U-Net)} & 59.17 & 72.39 & 54.05 & 60.86 & 61.62 \newline
> \text{LRM-SDXL} & \text{SDXL (U-Net)} & 60.35 & 71.19 & 53.80 & 61.58 & 61.73 \newline
> \hline
> \text{DiNa-LRM (SDXL)} & \text{SDXL (U-Net)} & 60.86 & 81.03 & 74.87 & 69.44 & 71.55 \newline
> \text{DiNa-LRM (SD3.5-M)} & \text{SD3.5-M (Dual-Stream-DiT)} & 60.34 & 82.13 & 75.04 & 68.43 & 71.49 \newline
> \text{DiNa-LRM (FLUX.1-Dev)} & \text{FLUX.1-Dev (Dual-Stream-DiT)} & 59.03 & 81.21 & 72.57 & 66.67 & 69.97 \newline
> \text{DiNa-LRM (Z-Image-Turbo)} & \text{Z-Image-Turbo (Single-Stream-DiT)} & 60.13 & 81.75 & 71.58 & 67.21 & 70.17 \newline
> \hline
> \end{array}
> $$
>
> These results show that the proposed formulation is **effective across different backbone families** and is **not specific to SD3.5-Medium alone.**
>
> ---
>
> `[W2-2] Transfer to other models sharing the same VAE.`
>
>
> We also add a transfer experiment across models **sharing the same latent space**. Specifically, we train the reward model on SD3.5-M and use it to align SD3.5-Large under the same ReFL setup. Following the protocol in our paper, we report both a held-out PickScore and the optimized proxy reward across training steps.
>
>
> **Transfer from SD3.5-M reward model to SD3.5-L alignment.**
>
> $$
> \begin{array}{lccccc}
> \hline
> \text{Metric} & \text{Step 0} & 50 & 100 & 150 & 200 \newline
> \hline
> \text{Held-out PickScore} & 20.80 & 20.92 & 21.08 & 21.27 & 21.52 \newline
> \text{Proxy Reward (DiNa-LRM)} & 0.24 & 0.51 & 0.73 & 1.09 & 1.29 \newline
> \hline
> \end{array}
> $$
>
> The held-out metric improves steadily throughout optimization, showing that a reward model trained on SD3.5-Medium can be effectively reused to **optimize a different generator instance within the same latent space**.
>
> ---
>
> `[W3] Narrow VLM comparison.`
>
> We appreciate this point. Our comparison in Table 1 focuses on **open, optimization-compatible reward models** that can serve as practical supervision signals in our alignment setting, including **CLIP-based rewards** (e.g., ImageReward, PickScore, HPSv2, MPS), **regression-based VLM rewards** (e.g., HPSv3), **generation-based VLM rewards** (e.g., UnifiedReward, UnifiedReward-Think), and prior **diffusion-based rewards** (e.g., LRM). This comparison scope is also consistent with prior open reward-model comparisons in this line of work[1,2].
>
> VLM-as-a-judge methods are important and we will expand the related-work discussion accordingly. At the same time, they are **not a directly matched comparison target for our setting**. They are typically **non-differentiable**, **API-based**, and are mainly used in offline preference optimization[3].
> We will revise the paper to make this scope and distinction clearer. Within this scope, DiNa-LRM remains a strong and substantially more efficient diffusion-native alternative to existing open, optimization-compatible VLM-based rewards.
>
> ---
>
> [1] Unified Multimodal Chain-of-Thought Reward Model through Reinforcement Fine-Tuning. NeurIPS 2025.
>
> [2] HPSv3: Towards Wide-Spectrum Human Preference Score. ICCV 2025.
>
> [3] Boost your own human image generation model via direct preference optimization with ai feedback. CVPR2025.

---

> > ### Author Rebuttal · Reviewer_QdW1 · 2026-04-02
> >
> > We thank the authors for their effort in preparing the rebuttal. However, after careful consideration, the responses do not fully resolve my concerns, and I maintain my score.
> >
> > **W1.** The rebuttal argues that the contribution lies in a "coherent formulation," but no new theoretical or experimental evidence beyond what was already in the submission has been provided. I acknowledge that the system integration is sound and that the empirical gains over prior diffusion-based baselines are meaningful, but the conceptual novelty remains insufficient, as noted in my original review.
> >
> > **W2.** I appreciate the pairwise accuracy evaluation across four backbones (SDXL, SD3.5-M, FLUX.1-Dev, Z-Image-Turbo). However, this table demonstrates that "the DiNa-LRM formulation can be independently applied to multiple backbones," which is distinct from "a single trained reward model can transfer to different backbones." For a method positioned as a general-purpose reward model, the latter is what matters. The SD3.5-M → SD3.5-L transfer experiment is noted, but this remains a transfer within the same model family and VAE, which is qualitatively different from cross-VAE transfer.
> >
> > **W3.** I understand the authors' clarification that the scope of comparison is limited to open, optimization-compatible reward models. However, the title "Beyond VLM-Based Rewards" sets an expectation of broadly superseding VLM-based reward modeling. As I noted in my original review, VLM-as-a-judge approaches represent a significant and growing branch of VLM-based reward modeling, and I find it difficult to support the title's claim without any comparison or substantive discussion of this paradigm.

---

> > > ### Author Response · Authors · 2026-04-03
> > >
> > > Thank you for the careful reading and the further response. We understand the remaining disagreement mainly concerns the intended scope of our claims, so we would like to make that scope explicit.
> > >
> > > ---
> > >
> > > **For W1.**
> > >
> > > We acknowledge that the broad idea of using diffusion backbones for reward modeling is not entirely unprecedented, and **the paper does not claim otherwise**. Our novelty claim is more specific. As discussed in Section 2, the closest prior diffusion-reward works[1, 2] already show that diffusion backbones can support reward learning on noisy states. However, their main focus is to make the reward model **usable** for step-level optimization, and **they largely carry over the clean-sample preference learning formulation** to the noisy-state setting without further investigating how it should be reformulated under diffusion noise.
> > >
> > > In contrast, our main novelty lies in **addressing this formulation gap**. we explicitly model the increased uncertainty of preference supervision under diffusion noise through a **noise-calibrated Thurstone formulation**. Building on this formulation, we further develop timestep-conditioned latent scoring and multi-noise inference ensembling. Therefore, the contribution of our work is better characterized as a **new diffusion-native preference formulation** for latent reward modeling, rather than a straightforward extension of prior latent reward methods. Empirically, our results show that this formulation consistently improves over prior diffusion-based reward baselines, including LRM, and our ablation studies further verify the effectiveness of the proposed design.
> > >
> > > [1] Diffusion Model as a Noise-Aware Latent Reward Model for Step-Level Preference Optimization. NeurIPS 2025.
> > >
> > > [2] Video generation models are good latent reward models. ArXiv 2025.11
> > >
> > > ---
> > >
> > > **For W2.**
> > >
> > > We believe the remaining concern here mainly stems from a different reading of “general-purpose” intended in the paper. Our intended claim is **not** that a single reward model trained on one backbone should zero-shot transfer across arbitrary backbones with different VAEs. Rather, “general-purpose” in our paper refers to a **reusable reward-model formulation** for preference alignment, as opposed to an algorithm-specific step-level reward tied to one optimization setting. To support this claim, we added two kinds of evidence in rebuttal: (i) applicability across four backbones (SDXL, SD3.5-M, FLUX.1-Dev, Z-Image-Turbo), and (ii) transfer within a shared latent space (SD3.5-M to SD3.5-L). Both results support the claimed generality of the proposed method.
> > >
> > > As for **cross-VAE transfer**, we view this as a broader challenge faced by latent reward modeling methods in general, as also discussed in Section 6 (Limitations), rather than a requirement specific to our method alone. Since the reward is defined on the latent space induced by a particular VAE, changing the VAE may substantially alter the geometry and semantics of that space. It is an important direction for future work, but it is beyond the scope of the present paper’s claims.
> > >
> > > ---
> > >
> > > **For W3.**
> > >
> > > We believe the remaining concern here is mainly about how the **title** is interpreted, rather than about the validity of the comparison setting itself. Our intended meaning of **“Beyond VLM-Based Rewards”** was not to claim that DiNa-LRM broadly **replaces** all paradigms of VLM-based reward modeling. Rather, our point is that preference alignment for diffusion models need not rely exclusively on VLM-based rewards, a diffusion-native latent reward model **can serve as a competitive and much more efficient alternative** within the reward-model setting studied in this paper.
> > >
> > > Accordingly, our experiments focus on **open, optimization-compatible reward models** that can act as practical supervision signals for alignment, which also follows the comparison protocol adopted in this line of work VLM-as-a-Judge approaches are an important related paradigm, and we will expand the discussion accordingly. However, they are not the directly matched baseline family for the setting considered here. We are glad to clarify or revise the wording in revision. In any case, this **does not affect the core contribution of the paper,** namely showing that a diffusion-native latent reward model can provide a strong and substantially more efficient alternative under the studied setting.
> > >
> > > ---
> > >
> > > Thank you again for your time and effort in reviewing our work.

---

### Official Review · Reviewer_GVfD · 2026-03-09

**Soundness:** 3
**Presentation:** 3
**Significance:** 3
**Originality:** 3
**Overall Recommendation:** 4
**Confidence:** 3

**Summary:**

This paper introduces DiNa-LRM, a novel approach to align diffusion models with human preferences directly in the latent space. Specifically, the authors introduce a "noise-calibrated Thurstone likelihood" that adapts traditional preference learning to noisy diffusion states by scaling the comparison uncertainty with the noise level. Built on a pretrained latent diffusion backbone, the model extracts multi-layer visual and textual features and aggregates them using a gated Q-Former and an MLP to compute a scalar reward. Extensive experiment are conducted to validate the effectiveness of DiNa-LRM.

**Compliance With Llm Reviewing Policy:**

Affirmed.

**Key Questions For Authors:**

- Why choosing a gated Q-Former over simpler alternatives to aggregate features? Is there any ablations studies on some vanilla architecture setting?

**Limitations:**

- Even though noisy ensemble shows promising results in pairwise accuracy benchmarking, its effectiveness in downstream RL is not validated.

- One of the biggest advantages of Diffusion Latent Reward is its use of **step wise RL**, whereas the current paper primarily evaluates it as a single terminal reward for pairwise accuracy. It would significantly strengthen the paper to demonstrate whether DiNa-LRM can provide dense, step-wise reward signals to improve trajectory-level RL optimization.

**Strengths And Weaknesses:**

- **Novelty**: It is the first paper introuducing **Thurstone likelihood in latent diffusion model** and adapt it to noisy diffusion states. By introducing a "noise-calibrated" comparison uncertainty that explicitly scales with the diffusion noise level, this approach elegantly handles the loss of semantic information at high noise levels.

- **Performance**: DiNa-LRM substantially outperforms existing diffusion-based reward baselines (such as LRM-SD1.5 & LRM-SDXL) across diverse benchmarks like ImageReward, HPDv2, HPDv3, and GenAI-Bench. Furthermore, it achieves pairwise preference accuracy that is strictly competitive with sota VLM-based reward but at a fraction of the computational cost.

---

> ### Author Rebuttal · Authors · 2026-03-31
>
> We thank the reviewer for the positive assessment of our paper, especially for recognizing the novelty of our proposed method, as well as the strong performance-efficiency tradeoff of DiNa-LRM. We address the questions below.
>
> ---
>
> `[Q1] Why gated Q-Former instead of simpler feature aggregation?`
>
> Thank you for this question. Our reward head needs to aggregate multi-layer visual features together with textual features, which benefits from **selective fusion rather than fixed pooling**. In addition, our inference-time multi-noise ensembling **concatenates features from multiple timesteps into a longer token context**, and the same query-based head is then used to aggregate them into a single score. This design is naturally supported by the Q-Former architecture.
>
> To validate this choice, we compare against two simpler alternatives:
> **(1) Average Pooling;**
> **(2) Transformer + Average Pooling.**
>
>
> **Reward-model evaluation (pairwise accuracy, %).**
>
> $$
> \begin{array}{lccccc}
> \hline
> \text{Method} & \text{ImageReward} & \text{HPDv2} & \text{HPDv3} & \text{GenAI Bench} & \text{Avg} \newline
> \hline
> \text{Average pooling} & 60.68 & 80.70 & 74.43 & 68.31 & 71.03 \newline
> \text{Transformer + Average pooling} & 60.98 & 80.71 & 74.70 & 68.18 & 71.14 \newline
> \textbf{Gated Q-Former (Ours)} & \mathbf{60.34} & \mathbf{82.13} & \mathbf{75.04} & \mathbf{68.43} & \mathbf{71.49} \newline
> \hline
> \end{array}
> $$
>
> All three variants are reasonably effective, while **gated Q-Former achieves the best average performance**. We will add this ablation to the revision.
>
> ---
>
> `[L1] Effectiveness of noisy ensembling in downstream RL optimization.`
>
> Following the reviewer’s suggestion, we further validate the effect of noisy ensembling in downstream alignment. Since **multi-noise ensembling is an inference-time, non-differentiable scoring strategy**, it cannot be directly used in reward-gradient-based optimization such as ReFL. We therefore evaluate it under **Flow-GRPO-Fast**[1], a fast-converging RL algorithm, on **SD3.5-M.** Following the evaluation protocol in our main paper, we**report held-out PickScore**, which is not used during optimization and therefore serves as a more reliable metric than raw reward values.
>
> **Effectiveness of noisy ensembling in RL optimization on SD3.5-M(held-out PickScore, optimized).**
>
> $$
> \begin{array}{lccccc}
> \hline
> \text{Method} & \text{Step 0} & 50 & 100 & 150 & 200 \newline
> \hline
> \text{DiNa-LRM + Flow-GRPO-Fast} & 19.62 & 21.22 & 21.28 & 21.61 & 21.88 \newline
> \textbf{DiNa-LRM(Ensemble) + Flow-GRPO-Fast} & \mathbf{19.62} & \mathbf{21.35} & \mathbf{21.56} & \mathbf{21.84} & \mathbf{22.03} \newline
> \hline
> \end{array}
> $$
>
> The results show that **noisy ensembling consistently improves the held-out metric across checkpoints**, making downstream optimization more stable and more effective.
>
> ---
>
>
> `[L2] Regarding step-wise preference optimization.`
>
> We appreciate this suggestion. In the current paper, we already formulate rewards on noisy states $r_\theta(x_t, t, c)$, which naturally supports reward modeling beyond terminal-only evaluation.
>
> To provide stronger evidence, we additionally evaluate DiNa-LRM in an LPO [2] setting. Briefly, LPO is a **step-level preference optimization method** that **performs optimization directly in the noisy latent space** on SDXL, using a latent reward model to provide preference signals at intermediate denoising steps rather than relying only on a terminal image-level reward.
> This makes it a suitable testbed for whether DiNa-LRM can serve as an effective reward signal for step-wise optimization. We report both a held-out PickScore and the optimized proxy reward across training epochs:
>
>
> **Step-wise Preferemce Optimization(LPO) on SDXL.**
>
> $$
> \begin{array}{lccccc}
> \hline
> \text{Metric} & \text{Epoch 0} & \text{Epoch 1} & \text{Epoch 2} & \text{Epoch 3} \newline
> \hline
> \text{Held-out PickScore} & 19.25 & 21.18 & 21.74 & 21.72 \newline
> \text{Proxy Reward(DiNa-LRM)} & -0.08 & 1.12 & 1.44 & 1.52 \newline
> \hline
> \end{array}
> $$
> Both the held-out metric and the proxy reward increase steadily throughout training.
> These results show that DiNa-LRM can indeed be **used as an effective reward signal in a step-level optimization setting**, rather than only as a terminal reward for pairwise evaluation.
>
> ---
>
> [1] Flow-GRPO: Training Flow Matching Models via Online RL. NeurIPS 2025.
>
> [2] Diffusion Model as a Noise-Aware Latent Reward Model for Step-Level Preference Optimization. NeurIPS 2025.

---

> > ### Author Rebuttal · Reviewer_GVfD · 2026-04-05
> >
> > Thanks for the author detailed ablation experiment, it resolve my problem and therefore i will keep my score.

---

> > > ### Author Response · Authors · 2026-04-07
> > >
> > > Thank you for your positive feedback and for your continued support. We sincerely appreciate the time and effort you took to read our rebuttal. We are glad that our response helped address your concerns. We are grateful for your positive assessment and continued support of our work.

---

### Official Review · Reviewer_FG8C · 2026-03-12

**Soundness:** 3
**Presentation:** 3
**Significance:** 2
**Originality:** 3
**Overall Recommendation:** 4
**Confidence:** 4

**Summary:**

This paper proposes DiNa-LRM, a diffusion-native latent reward model for preference optimization of diffusion models, which directly leverages the latent representations inside diffusion models instead of relying on VLM-based rewards. The method learns rewards on noisy latent states and adjusts comparison uncertainty according to the noise level via a noise-calibrated Thurstone likelihood, enabling more consistent preference learning. DiNa-LRM substantially outperforms existing diffusion-based reward models and achieves competitive accuracy compared to VLM-based rewards, while significantly reducing memory usage and computational cost.

**Compliance With Llm Reviewing Policy:**

Affirmed.

**Final Justification:**

The paper is technically sound and practically relevant, with clear efficiency gains and strong results over prior diffusion-based baselines. My main concerns were largely addressed in the rebuttal through added comparisons with LRM-SDXL, longer-horizon and alternative-algorithm validation, and broader backbone evaluations. I therefore raise my score and support acceptance.

**Key Questions For Authors:**

In Table 1, LRM-SDXL is used as a primary baseline for reward model evaluation, and the difference from the proposed method is clearly shown. On the other hand, in the ReFL experiment in Section 5.4, LRM-SDXL is not included as a comparison target. LRM-SDXL was originally designed primarily for step-level preference optimization and is the most closely related existing diffusion-native reward model to the proposed method in terms of reward assignment on noisy latent states. For this reason, I consider the comparison to be particularly important in the alignment task.

Could you explain the technical reasons for not including LRM-SDXL as a comparison target in the ReFL experiment? If LRM-SDXL can work without issues under the ReFL setting, adding this comparison result would be essential for demonstrating the superiority of the proposed method in alignment more convincingly. Conversely, if there are technically valid reasons, providing that explanation would significantly alleviate concerns about the adequacy of the current experimental design.

**Limitations:**

yes

**Strengths And Weaknesses:**

### Strengths

- This paper addresses an important topic: using diffusion models themselves as reward models. While prior work exists in this area, this paper focuses on repurposing diffusion backbones as general-purpose reward models that can be used in the same scenario as VLM rewards. From a practical perspective, using the diffusion model itself as the reward backbone aligns the representation space between the generator and the reward model, which can provide a more consistent evaluation signal. It also offers advantages in terms of memory usage and computational cost. Therefore, this topic occupies an important position in post-training alignment of diffusion models.

- A clear strength of this paper is that, at least within the scope of reward model evaluation, the proposed method shows consistent improvements over the existing diffusion-based reward baseline (LRM), and its effectiveness is demonstrated relatively clearly. In particular, rather than only reporting final performance, the authors conduct systematic ablation studies on major design choices, including training-time noise conditions, variance modeling, multi-noise inference ensembling, and backbone adaptation strategies, confirming the contribution of each component individually. This is appreciated.

### Weaknesses

- In the ReFL experiment, while the effectiveness of the proposed method as a supervisory signal is demonstrated, the most closely related existing diffusion-native reward, LRM-SDXL, is not included as a comparison target, which raises concerns about the completeness of the comparison. In particular, in Table 1, LRM-SDXL is treated as a primary baseline and the superiority of the proposed method is claimed based on reward model evaluation alone. However, LRM-SDXL was not originally proposed as a general-purpose final reward model, but was designed primarily for step-level preference optimization on noisy latent states. That said, given that LRM-SDXL is the most closely related diffusion-native reward model, it remains unclear why it was not evaluated as a comparison target in the alignment task. At the very least, an explicit explanation of the technical reasons for not including LRM-SDXL in the ReFL experiment is needed.

- Furthermore, the preference alignment experiment is mainly limited to a 150-step comparison using ReFL, and the paper only states that "no obvious evidence of reward hacking at the early iterations" is observed. Conversely, it remains unknown whether the method is stable under long-horizon optimization or with different alignment algorithms. The robustness of reward optimization does not appear to be clearly demonstrated.

- The effectiveness of the proposed method is primarily validated on a single diffusion backbone, SD3.5-Medium, and the evidence appears insufficient to judge the generality of the approach. The authors themselves acknowledge that the reward model is learned and evaluated in "the latent space of a specific diffusion backbone" and that "cross-backbone transfer is not guaranteed." Therefore, it is currently not possible to disentangle whether the observed improvements stem from the intrinsic effectiveness of diffusion-native reward modeling itself, or whether they are dependent on the representation space or architecture of this particular backbone. Without additional validation across multiple backbones or model families, the claims regarding the generality of the method remain somewhat limited.

- There are several typos. Figure 1: "Fieldity Loss" should be "Fidelity Loss". Figure 4: "PEAK VARM" should be "PEAK VRAM". The beginning of Section 6 (Conclusion): "Throughtout" should be "Throughout".

---

> ### Author Rebuttal · Authors · 2026-03-31
>
> We thank the reviewer for the careful reading and for recognizing the importance of diffusion-native reward modeling, the practical efficiency benefits, and the systematic ablations in our paper. We address the concerns below.
>
> ---
>
> `[W1 & Q1] Comparison with LRM-SDXL in alignment.`
>
> Thank you for highlighting this important point. In the original submission, our alignment experiments were conducted only on the SD3.5-M backbone, whereas LRM-SDXL[1] is tied to SDXL backbone. To enable a fair comparison, we additionally **trained DiNa-LRM on SDXL and compared it against LRM-SDXL in two settings**:
>
>
> **(1) Reward-model evaluation (pairwise accuracy, %).**
>
> DiNa-LRM (SDXL) consistently outperforms LRM-SDXL across datasets.:
>
> $$
> \begin{array}{lccccc}
> \hline
> \text{Method} & \text{ImageReward} & \text{HPDv2} & \text{HPDv3} & \text{GenAI Bench} & \text{Avg} \newline
> \hline
> \text{LRM-SDXL} & 60.35 & 71.19 & 53.80 & 61.58 & 61.73 \newline
> \textbf{DiNa-LRM (SDXL)} & \mathbf{60.86} & \mathbf{81.03} & \mathbf{74.87} & \mathbf{69.44} & \mathbf{71.55} \newline
> \hline
> \end{array}
> $$
>
>
> **(2). ReFL alignment on SDXL (held-out PickScore).**
>
> Following the evaluation protocol in our main paper, we report **held-out PickScore**, which is not used for training and thus serves as a more reliable alignment metric than raw reward values. DiNa-LRM shows stronger gains:
>
> $$
> \begin{array}{lccccc}
> \hline
> \text{Method} & \text{Step 0} & 50 & 100 & 150 & 200 \newline
> \hline
> \text{LRM-SDXL + ReFL} & 19.51 & 19.89 & 20.32 & 21.30 & 21.43 \newline
> \textbf{DiNa-LRM (SDXL) + ReFL} & \mathbf{19.51} & \mathbf{19.85} & \mathbf{20.99} & \mathbf{21.59} & \mathbf{21.99} \newline
> \hline
> \end{array}
> $$
>
> ---
>
> `[W2-1] Long-horizon optimization robustness.`
>
> Thank you for raising this point. ReFL[2] is itself a fast-converging alignment method, and the original ReFL paper also focuses on **relatively short optimization horizon(~100 steps)** with pretrained-loss regularization. Our original 150-step setting mainly follows this regime, and as stated in Appendix B.3, **we did not include extra regularization** in the initial experiment.
>
> We further tested a longer-horizon setting with pretrained-loss regularization. The held-out PickScore improves steadily from 0 to 600 steps, and then declines afterward:
>
> $$
> \begin{array}{lcccccc}
> \hline
> \text{Method} & \text{Step 0} & 100 & 200 & 400 & 600 & 800 & 1000 \newline
> \hline
> \text{DiNa-LRM + ReFL + Reg.} & 19.62 & 20.39 & 20.76 & 21.47 & 21.91 & 20.01 & 18.74 \newline
> \hline
> \end{array}
> $$
>
> This suggests that **DiNa-LRM remains effective over substantially longer horizons when regularized**, although reward hacking can still emerge under very long optimization, which is a common challenge in reward-based alignment. We will include these results and clarify this discussion in the revision.
>
> ---
>
> `[W2-2] Different alignment algorithms.`
>
> Beyond ReFL, we also evaluated DiNa-LRM under **Flow-GRPO-Fast**[3], a fast-converging RL algorithm that typically shows clear gains within hundreds of steps. We ran this experiment on SD3.5-M, and the held-out PickScore improves consistently, suggesting that **DiNa-LRM is also effective beyond the ReFL setting**. We will include these results in the revision.
>
>
> $$
> \begin{array}{lccccc}
> \hline
> \text{Method} & \text{Step 0} & 50 & 100 & 150 & 200 \newline
> \hline
> \text{DiNa-LRM + Flow-GRPO-Fast} & 19.62 & 21.22 & 21.28 & 21.61 & 21.88  \newline
> \hline
> \end{array}
> $$
>
> ---
>
> `[W3] Generality across backbones / model families.`
>
> Thank you for this important question. To strengthen the evidence on generality, we additionally conducted two types of experiments:
>
> **(1) Across different model families**, including UNet (SDXL), single-stream DiT (Z-Image), and dual-stream DiT (SD3.5 / FLUX);
>
> **(2) Transfer across models sharing the same latent space**, where a reward model trained on SD3.5-M is used to align SD3.5-L.
>
> As shown in our responses to `Reviewer #QdW1 [W2-1] & [W2-2]`, both settings show positive results, suggesting that **the benefit is not specific to SD3.5-Medium alone** and that **transfer is possible when the latent space is shared**.
>
> ---
>
> `[W4] Typos.`
>
> Thank you for your careful reading and for catching these mistakes. We will correct them in the revision.
>
> ---
>
> [1] Diffusion Model as a Noise-Aware Latent Reward Model for Step-Level Preference Optimization. NeurIPS 2025.
>
> [2] ImageReward: Learning and Evaluating Human Preferences for Text-to-image Generation. NeurIPS 2023.
>
> [3] Flow-GRPO: Training Flow Matching Models via Online RL. NeurIPS 2025.

---

> > ### Author Rebuttal · Reviewer_FG8C · 2026-04-02
> >
> > Thank you for the careful rebuttal. I believe that, overall, my concerns have been largely addressed. In particular, the additional comparison with LRM-SDXL, the validation under longer-horizon optimization and a different alignment algorithm, and the added evaluations across multiple backbones and within a shared latent space have substantially alleviated my concerns about the adequacy of the experimental design and the generality of the method. Although some limitations remain, I now consider the paper worthy of acceptance and am raising my score.

---

> > > ### Author Response · Authors · 2026-04-02
> > >
> > > Thank you for your careful reconsideration and for your support. We sincerely appreciate the time and effort you took to read our rebuttal. We are glad that our response helped address your concerns, including the experimental design and the generality of the method. We will make sure these improvements are clearly reflected in the final version.

---

### Decision · Program_Chairs · 2026-04-30

**Decision:**

Accept (regular)

**Comment:**

Most reviewers have a fairly positive view of this paper.  The basic idea of using Diffusion models for reward modeling in conjunction with generation is not entirely new, but reviewers found the particulars compelling and persuasive. The experiments were generally considered robust. Reviewer QdW1 is an outlier, his concerns were about novelty, which I think is partially applicable  since joint diffusion+reward training has been seen before, but the approach here is new (as other reviewers acknowledge). Other complaints seem to focus on the specifics of the claims made, which authors have acknowledged in the last round and will tidy up their language.